# *Drosophila* TMEM63 and mouse TMEM63A are lysosomal mechanosensory ion channels

Kai Li[1,2,4], Yanmeng Guo[1,2,4], Yayu Wang[1,2,4], Ruijun Zhu[1,2], Wei Chen [1,2], Tong Cheng[1,2], Xiaofan Zhang[1,2], Yinjun Jia[3], Ting Liu[3], Wei Zhang [3], Lily Yeh Jan [1,2] & Yuh Nung Jan [1,2] ✉

Cells sense physical forces and convert them into electrical or chemical signals, a process known as mechanotransduction. Whereas extensive studies focus on mechanotransduction at the plasma membrane, little is known about whether and how intracellular organelles sense mechanical force and the physiological functions of organellar mechanosensing. Here we identify the *Drosophila* TMEM63 (*Dm*TMEM63) ion channel as an intrinsic mechanosensor of the lysosome, a major degradative organelle. Endogenous *Dm*TMEM63 proteins localize to lysosomes, mediate lysosomal mechanosensitivity and modulate lysosomal morphology and function. *Tmem63* mutant flies exhibit impaired lysosomal degradation, synaptic loss, progressive motor deficits and early death, with some of these mutant phenotypes recapitulating symptoms of TMEM63-associated human diseases. Importantly, mouse TMEM63A mediates lysosomal mechanosensitivity in Neuro-2a cells, indicative of functional conservation in mammals. Our findings reveal *Dm*TMEM63 channel function in lysosomes and its physiological roles in vivo and provide a molecular basis to explore the mechanosensitive process in subcellular organelles.

The eukaryotic cell has many intracellular compartments, such as membrane-bound organelles, non-membranous structures and cytoskeleton, in a dynamic cytoplasmic environment. The membranes of intracellular organelles are frequently faced with mechanical challenges, especially in the following biological contexts. Firstly, cytoskeletal elements directly exert force on organelles through actomyosin contraction[1], cytoskeleton dynamics[2] or motor-protein-based transport[3]. Secondly, the intertwined trafficking of organelles in the limited cytoplasmic space results in physical contacts between organelles[4]. Moreover, most organelles are highly dynamic structures. The fast remodelling of organelles, exemplified in fusion, fission and tubulation, involves changes in membrane curvature and surface tension[5,6]. Recent studies have demonstrated the important roles of intracellular forces in regulating mitochondrial fission[7], vesicle fusion[8], lysosomal positioning[9] and nucleocytoplasmic transport[10]. However, whether

organelles sense mechanical forces and, if so, how the force is sensed by an organelle remains largely unknown.

At the plasma membrane, force sensing is mediated by mechanosensitive (MS) ion channels[11], as well as curvature-sensing proteins[12]. MS channels convert mechanical stimuli to ion flux across the membrane, allowing a very fast transduction of mechanical stimuli to yield electrical signals. Besides activating MS channels, mechanical forces induce conformational changes of certain curvature-sensing proteins to initiate downstream biochemical signalling cascades. Apart from the nucleus, which is MS and responds to force through curvature-sensing proteins located in the nuclear envelope[13], the mechanical properties of other organelles are largely unknown.

The lysosome, a membrane-bound organelle responsible for the degradation of intracellular materials, is essential to control basic cellular functions such as nutrient sensing, metabolic adaptation and

[1]Department of Physiology, University of California at San Francisco, San Francisco, CA, USA. [2]Howard Hughes Medical Institute, University of California at San Francisco, San Francisco, CA, USA. [3]School of Life Sciences, Tsinghua-Peking Center for Life Sciences, IDG/McGovern Institute for Brain Research, Tsinghua University, Beijing, China. [4]These authors contributed equally: Kai Li, Yanmeng Guo, Yayu Wang. ✉e-mail: yuhnung.jan@ucsf.edu

cell growth and survival[14,15]. Ion flux across the lysosomal membrane plays an important role to maintain ion homeostasis and lysosomal function[16]. Several lysosomal ion channels have been discovered and dysfunction of lysosomal channels is associated with lysosomal storage diseases[17], immunodeficiency disorders[18,19] and neurodegenerative diseases[20]. Lysosomes undergo frequent membrane remodelling through fusion[21], fission[22] and tubulation[23] processes involving changes of the lysosomal surface-to-volume ratio and membrane curvature. Such alternation in membrane curvature leads to a change in membrane tension that may be sensed by putative MS proteins in the lysosomal membrane. Indeed, yeast and plant vacuoles, the functional equivalent of lysosomes in animal cells, exhibit MS Ca²⁺ efflux during vacuolar remodelling[24,25]. These phenomena raise the question of whether MS channels are present in lysosomes of animal cells. In this Article, we report that *Drosophila* fat-body lysosomes are intrinsically MS. We identified *Drosophila* TMEM63 (*Dm*TMEM63) as a MS channel to mediate lysosomal mechanosensitivity and to regulate lysosomal morphology and function in vivo, demonstrating an important role of mechanosensing in lysosomal homeostasis.

## Results

### *Dm*TMEM63 localizes to lysosomes

To identify organellar mechanosensors, we examined subcellular localization patterns of several established or putative MS ion channels including *Drosophila* Piezo (*Dm*Piezo)[26], NompC[27], Iav[28] and *Dm*TMEM63 (ref. 29). We expressed each channel protein in *Drosophila* Schneider 2 (S2) cells and tested for its co-localization with organellar markers including LAMP1 (late endosomes and lysosomes), Mito (mitochondria) and SKL (peroxisomes). These MS channel proteins showed distinct subcellular distribution patterns. *Dm*Piezo proteins were present primarily on the plasma membrane (Extended Data Fig. 1a). NompC and Iav proteins showed intracellular localization, but neither of these channels co-localized with LAMP1, Mito or SKL (Extended Data Fig. 1b,c). In contrast, *Dm*TMEM63 proteins were prominent in the intracellular compartments positive for LAMP1, but not Mito or SKL (Fig. 1a,b). Thus, among the MS channels we screened, *Dm*TMEM63 specifically localized in the late endosomes and lysosomes (for simplicity, referred to as lysosomes hereafter).

Next, we examined the localization of endogenous *Dm*TMEM63 proteins in vivo. We first investigated the tissue expression patterns of the *Tmem63* gene in *Drosophila* using the *Tmem63*-Gal4 reporter line and found broad expression in many tissues (Extended Data Fig. 2a). We then corroborated the broad tissue expression profile of *Tmem63* transcript through analysis of RNA sequencing datasets (Extended Data Fig. 2b). Next, to visualize the subcellular localization of endogenous *Dm*TMEM63 proteins, we inserted green fluorescent protein (GFP) or mCherry into the *Tmem63* native locus to generate *DmTmem63*^GFP and *DmTmem63*^mCherry lines with the fluorescent protein fused to the C-terminus of the endogenous *Dm*TMEM63 (Fig. 1c). The precise integration into the *Drosophila* genome was confirmed by DNA sequencing and the full-length fusion proteins were detected by immunoblots (Fig. 1d). We found abundant expression of endogenous *Dm*TMEM63 proteins in intracellular vesicles positive for the lysosomal marker LAMP1 in fat-body cells or the lysosomal marker Spin in ventral nerve cord (VNC) neurons (Fig. 1e). We further showed that the lumen of the vesicles containing *Dm*TMEM63 in the fat body displayed LysoTracker staining (Fig. 1f), suggesting these vesicles are organelles with acidic lumen. Notably, the pH-sensitive GFP showed fluorescence in *DmTmem63*^GFP flies (Fig. 1f), which suggests the C-terminus of the *Dm*TMEM63 faces to the cytosol, not the acidic lumen. In addition, the membrane-located patterns of *Dm*TMEM63 (Fig. 1e,f) demonstrated *Dm*TMEM63 is a lysosome-resident protein rather than a cargo that is to be degraded in the lysosomal lumen. Taken together, these results indicate that *Dm*TMEM63 proteins are localized to lysosomes.

### *Dm*TMEM63 is a mechano- and osmo-sensitive channel

The unresponsiveness to agents for lysosomal enlargement hinders recordings of *Dm*TMEM63 currents from the lysosomal membrane in S2 cells[20]. Because the fluorescent signals, although relatively weak, were also observed on the cell surface of S2 cells overexpressing *Dm*TMEM63–GFP (Fig. 1a), we characterized the electrophysiological properties of plasma membrane located *Dm*TMEM63 in these cells. We applied negative pressure on the plasma membrane of the S2 cells (Fig. 2a) and found that *Dm*TMEM63–GFP-expressing cells, but not GFP-expressing cells, showed currents in response to mechanical stimuli (Fig. 2a). The current amplitude progressively increased as the pressure gradually strengthened (Fig. 2b). The half-activation pressure of *Dm*TMEM63 was substantially higher than that of NompC (Extended Data Fig. 3a,b) and the single-channel conductance of *Dm*TMEM63 was very small (Extended Data Fig. 3c), in line with previous reports that TMEM63 proteins are high-threshold MS channels with small conductance[29,30]. Furthermore, we found that the *Dm*TMEM63 channel was permeable to Na⁺ and K⁺ (Fig. 2c), but not the larger cation *N*-methyl-D-glucamine (NMDG) (Fig. 2c), suggesting that the *Dm*TMEM63 channel conducts cations, with its ion permeation property similar to that of its mammalian homologues[29].

Hypo-osmotic stress, a physical stimulus involving mechanical force, can activate the mammalian TMEM63 channels[31]. To test whether *Dm*TMEM63 is responsive to hypotonic treatment, we performed calcium imaging experiments to monitor *Dm*TMEM63-mediated Ca²⁺ signals. As illustrated in Fig. 2d, we fused GCaMP6f with the *Dm*TMEM63 C-terminus, which is exposed to the cytosol and in proximity with the channel pore, to ensure the *Dm*TMEM63-conducted lysosomal Ca²⁺ efflux can be readily detected before diffusion and dilution in the cytosol. With hypotonic treatment, we found dramatic Ca²⁺ spikes in S2 cells expressing *Dm*TMEM63-fused GCaMP6f (Fig. 2e–h), but not in cells expressing LAMP1-fused GCaMP6f (Fig. 2e–g), a control construct used to monitor the basal level of lysosomal Ca²⁺ signals. With the Ca²⁺-free hypotonic solution used in this experiment, there was no Ca²⁺ influx into the cell. Thus, the Ca²⁺ signals observed in *Dm*TMEM63-expressing cells are mainly derived from *Dm*TMEM63-localized lysosomes. Together, these results demonstrate that the *Dm*TMEM63 channels at lysosomes are permeable to calcium ions and can be activated by a hypo-osmotic shock.

### *Dm*TMEM63 mediates lysosomal mechanosensitivity

To investigate the functional roles of the *Dm*TMEM63 channel, we generated two knockout (KO) alleles. For the *Tmem63*^1 mutant, the entire coding sequence of the *Tmem63* gene was removed (Extended Data Fig. 4a,b). For the *Tmem63*^2 mutant, the coding sequence was replaced with Gal4 (Extended Data Fig. 4c,d). The genomic deletions of the two alleles were confirmed through genotyping followed by DNA sequencing.

The mechanosensitivity and lysosomal localization of *Dm*TMEM63 channels raised intriguing questions regarding the physiological role of *Dm*TMEM63 in the lysosomal membrane. To investigate whether the lysosome responds to mechanical pressure, we performed patch-clamp recordings on native lysosomal membranes. As illustrated in Fig. 3a, the larval fat bodies were dissected and ruptured, and their lysosomes were released and placed on the coverslip for patch-clamp recording. The intact lysosomes appeared as GFP–LAMP1-positive vesicles (Fig. 3b,c). We applied mechanical pressure to stretch the lysosomal membrane and observed MS currents (Fig. 3d). The current amplitude progressively increased as the pressure level was raised (Fig. 3d,e). Furthermore, the MS currents of lysosomes were largely abolished in *Tmem63* mutants (Fig. 3d,f). Thus, we demonstrate that the *Drosophila* fat-body lysosome is a MS organelle, and its mechanosensitivity requires the *Dm*TMEM63 channel.

We also performed patch-clamp recordings to examine whether there is a native *Dm*TMEM63 currents in the plasma membranes of the fat body. In contrast to the apparent MS currents recorded from

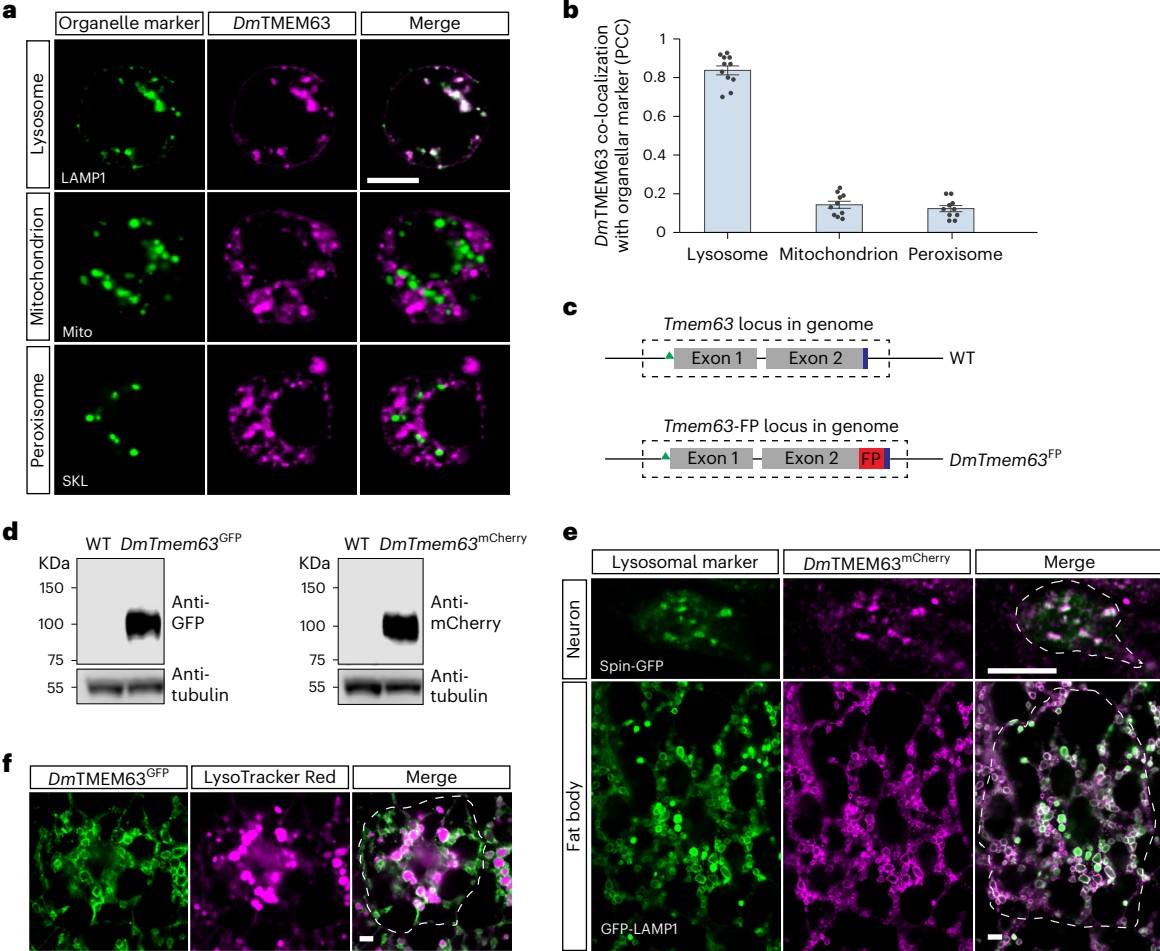

**Fig. 1 | *Dm*TMEM63 localizes to lysosomes. a**, Representative images showing the expression patterns of the *Dm*TMEM63 proteins and organellar markers in S2 cells. **b**, Quantification of the co-localization between *Dm*TMEM63 and lysosome (*n* = 11 cells), mitochondrion (*n* = 10 cells) or peroxisome (*n* = 10 cells). Data shown are mean ± s.e.m. **c**, Schematic representation of the genomic locus of *Tmem63* in WT and *DmTmem63*^FP (FP, fluorescent protein) knock-in flies. GFP or mCherry was inserted to follow the last amino acid of the endogenous *Dm*TMEM63 protein. The green triangle and blue bar indicate the start codon and stop codon, respectively. **d**, Immunoblots of head extracts in WT, *DmTmem63*^GFP and *DmTmem63*^mCherry flies showing the presence of full-length fusion proteins.

**e**, Co-localization of *Dm*TMEM63 proteins with lysosomal markers in VNC motor neurons and fat bodies of the wandering-stage *DmTmem63*^mCherry fly larvae. Lysosomes were labelled by GMR51B08-Gal4-driven Spin–GFP or Cg-Gal4-driven GFP–LAMP1. **f**, Representative images showing the expression of *Dm*TMEM63 proteins and LysoTracker-Red signals in the fat body of the *DmTmem63*^GFP wandering-stage larvae. LysoTracker-stained acidic organelles. For **d**–**f**, the results are representative of three independent experiments. For **e** and **f**, dashed lines indicate a single cell within the tissues. Scale bars, 5 μm (**a**,**e** and **f**). Numerical data and unprocessed blots are available as Source data.

fat-body lysosomes, we did not detect MS currents from the fat-body plasma membrane (Extended Data Fig. 5). The single-channel currents observed in the wild-type (WT) cells were not abolished in the *Tmem63* mutant cells (Extended Data Fig. 5), suggesting these currents are not conducted by *Dm*TMEM63. Together, these results suggest that the endogenous *Dm*TMEM63 channel mainly functions at lysosomes, but not the plasma membrane, of the fat-body cells.

### *Dm*TMEM63 modulates lysosomal morphology

Lysosomes undergo frequent morphological remodelling that involves changes in membrane curvature and tension. Having identified *Dm*TMEM63 as a lysosomal mechanosensor, we wondered whether changes in local mechanics of lysosomal membrane may activate the *Dm*TMEM63 channel in the physiological conditions. To approach this question, we evaluated the correlation between lysosomal morphology and the *Dm*TMEM63-mediated Ca²⁺ signal. As shown in Extended Data Fig. 6a, we transfected S2 cells with a tandem fluorescence-tagged *Dm*TMEM63 (*Dm*TMEM63–mCherry–GCaMP6f), which enabled us to simultaneously visualize lysosomal morphological changes

and monitor *Dm*TMEM63-mediated Ca²⁺ flux. We found a positive correlation between the lysosomal membrane curvature and the lysosomal Ca²⁺ signals (Extended Data Fig. 6b,c), indicating the *Dm*TMEM63 channel activity may vary with the morphological dynamics of lysosomes.

We further investigated the role of *Dm*TMEM63 in the morphological remodelling of lysosomes in vivo. *Drosophila* fat body, which is equivalent to the vertebrate liver and adipose tissue, has been widely used as a model system to study autophagy and lysosomal remodelling[32,33]. The fat-body lysosome is highly dynamic and adaptive to nutrient availability[32,33]. To test whether *Dm*TMEM63 functions in lysosomal remodelling, we labelled the fat-body lysosomes using GFP–LAMP1 and examined the lysosomal morphology of the WT and *Tmem63* mutant cells in both fed and starved conditions. Under the fed condition, the lysosomal size in *Tmem63* mutant cells was slightly increased compared with that in WT cells (Fig. 4a,b). Under the starvation condition in which lysosomal maturation is promoted and lysosomal membrane dynamics is more active[32,33], the size of lysosomes increased in cells of both genotypes (Fig. 4c), and the lysosomes in *Tmem63* mutant cells

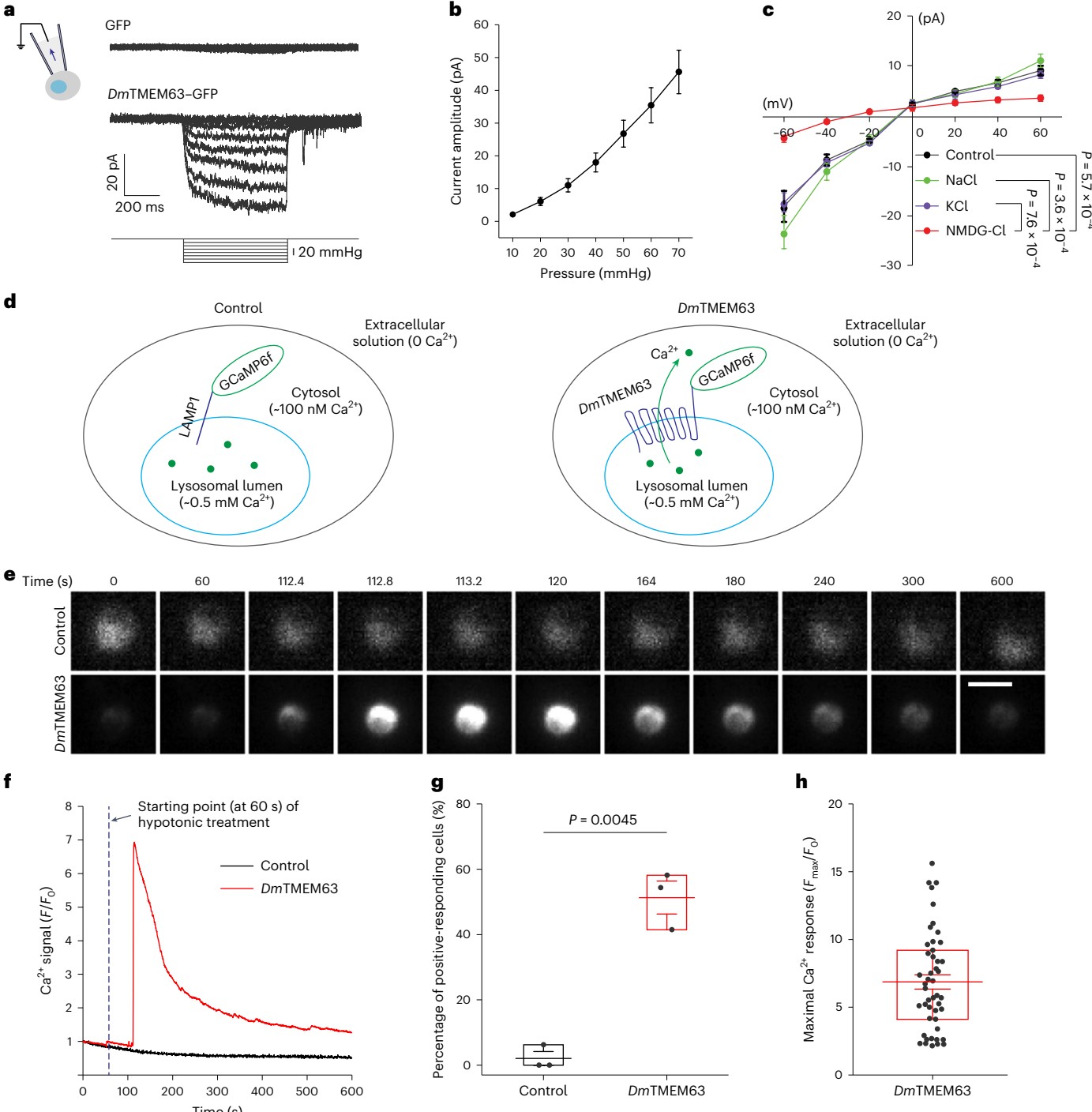

**Fig. 2 | *Dm*TMEM63 is a mechano- and osmo-sensitive channel.**
**a**, Representative MS current traces recorded from S2 cells expressing GFP or
*Dm*TMEM63–GFP at a holding potential of −80 mV. **b**, Dose-dependence curve
of stretch-induced currents. $n = 9$ cells. **c**, *I*–*V* curves of MS currents recorded
in S2 cells expressing *Dm*TMEM63–GFP. Currents were evoked by pressure
at −45 mmHg. Control ($n = 11$ cells) is the standard pipette solution, and NaCl
($n = 13$ cells), KCl ($n = 10$ cells) and NMDG-Cl ($n = 7$ cells) indicate pipette solution
consisting NaCl, KCl and NMDG-Cl, respectively (Methods). **d**, A schematic
showing the strategy of lysosomal calcium imaging. S2 cells expressing
GCaMP6f-fused LAMP1 (control group) or GCaMP6f-fused *Dm*TMEM63
(*Dm*TMEM63 group) were perfused with $Ca^{2+}$-free hypotonic solution
(160 mOsm $l^{-1}$) revealing $Ca^{2+}$ efflux from lysosomes. The estimated
$Ca^{2+}$ concentrations[16] in the lysosomal lumen and cytosol are indicated.

**e**,**f**, Representative time-lapse images (**e**) and $Ca^{2+}$ intensity traces (**f**) showing
the GCaMP6f signals in response to hypotonic treatment. Scale bar, 10 μm.
**g**, Percentage of cells responsive to hypo-osmolarity in control ($n = 34$ cells) or
*Dm*TMEM63 ($n = 89$ cells). $Ca^{2+}$ intensity increases by over 70% were considered
as positive-responding cells in three independent experiments. **h**, The maximal
$Ca^{2+}$ signals of positive-responding cells in the *Dm*TMEM63 group. $n = 47$
cells. Data shown are mean ± s.e.m. (**b**,**c**). In **c**, the statistical significance was
determined by two-sided Mann–Whitney test and the *P* values at −60 mV are
shown. In **g**, the statistical significance was determined by unpaired two-sided
Student's *t*-test. In **g** and **h**, lines represent the mean, bounds of box edge
the interquartile range, and whiskers indicate the range of standard error.
Numerical data are available as Source data.

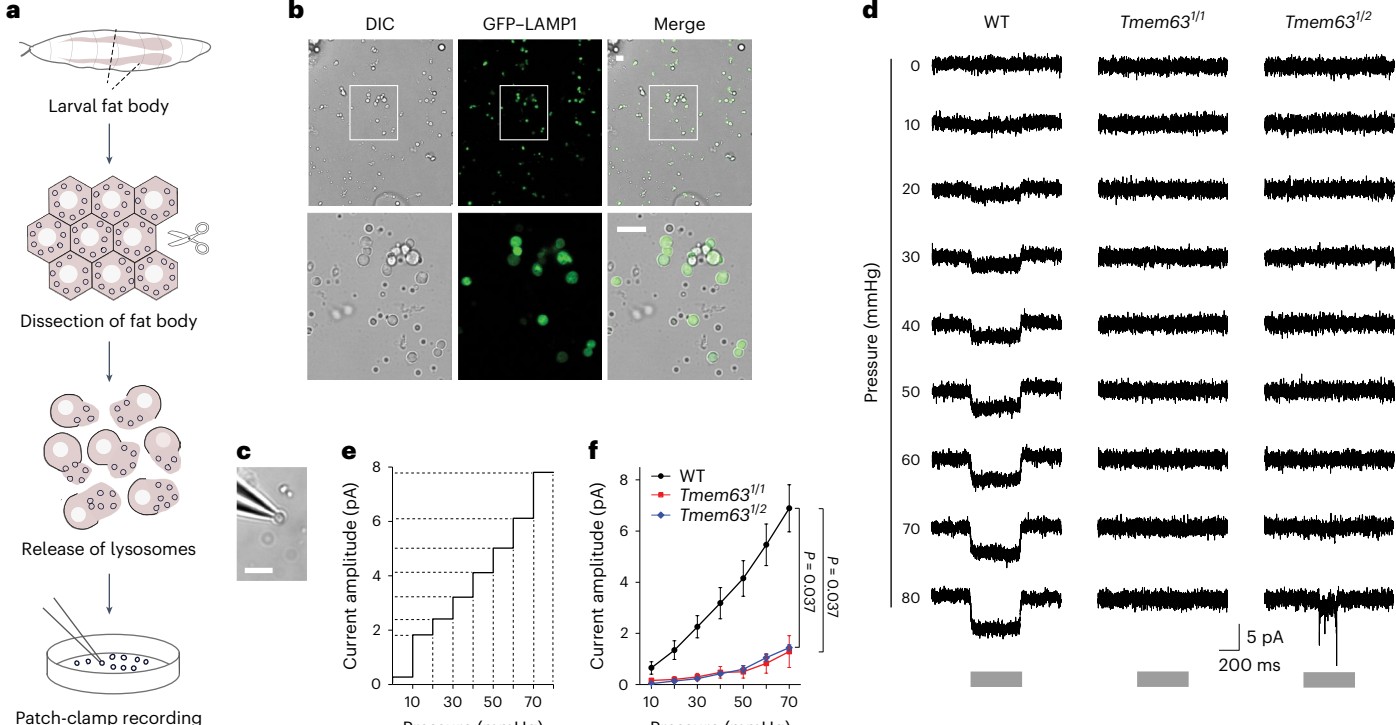

**Fig. 3 | *Dm*TMEM63 mediates lysosomal mechanosensitivity. a**, Schematic of the experimental procedure for patch-clamp recording from native lysosomes of the *Drosophila* fat body. Fat bodies were dissected from late-L3 stage larvae. Lysosomes were released from broken fat body cells and placed on the coverslips for recording. **b**, Top: images showing the lysosomes (GFP–LAMP1-positive vesicles) on the recording coverslip. Bottom: a magnified view of the boxed region. DIC, differential interference contrast. **c**, An image showing a patch-clamped lysosome on the tip of a recording pipette. **d**, MS currents of the lysosomal membranes in response to negative pressure in lysosome-attached

patch-clamp configuration. Lysosomes were isolated from WT flies and *Tmem63* mutants, respectively. Pressures were applied from −10 to −80 mmHg (10 mmHg per step) at a holding potential of −60 mV. Grey bars indicate the time duration of the pressure clamp. **e**, A surface plot showing the current responses of a single lysosomal patch. **f**, Group data of the dose-dependence curve of the MS currents. (WT, *n* = 6 lysosomes; *Tmem63^{1/1}*, *n* = 3 lysosomes; *Tmem63^{1/2}*, *n* = 3 lysosomes). Data shown are mean ± s.e.m. The statistical significance was determined by two-sided Mann–Whitney test, and the *P* values at −70 mmHg were shown. Scale bars, 5 μm (**b** and **c**). Numerical data are available as Source data.

---

were markedly larger than those in WT cells (Fig. 4c,d). The increased size of lysosomes in *Tmem63* mutant cells was also confirmed through LysoTracker staining (Extended Data Fig. 7a,b).

We further investigated the effect of an increase in the *Dm*TMEM63 level on lysosomal morphology. We observed a dramatic change in the distribution and shape of the lysosomes in *Tmem63*-overexpression (*Tmem63*-OE) cells. Under the fed condition, *Tmem63*-OE cells had lysosomes positioned close to the periphery, in contrast to the evenly distributed lysosomes in control cells (Fig. 4e,f). Under the starvation condition, lysosomes in *Tmem63*-OE cells exhibited not only peripheral positioning (Fig. 4g,h) but also an irregular shape (Fig. 4g,h). Thus, alternation of *Dm*TMEM63 expression dramatically remodels lysosomal morphology.

**Reduced lifespan and age-dependent motor deficits in *Tmem63* mutants**

The *Tmem63* mutant flies were viable and fertile and showed apparently normal development to the adult stage (Fig. 5a–c). However, we observed a reduced lifespan for the *Tmem63* mutant flies (Fig. 5d). Furthermore, adult *Tmem63* mutant flies exhibited age-dependent motor deficits as evident from the negative geotaxis assay. WT flies performed well in climbing at day 4 and day 16, and the performance declined at day 30 (Fig. 5e). In contrast, whereas *Tmem63* mutant flies performed well at day 4, they showed decline in climbing index by nearly 40% at day 16 compared with their WT counterparts (Fig. 5e). By day 30, the climbing abilities of the *Tmem63* mutants were almost eliminated (Fig. 5e). These behavioural defects could be rescued by re-expressing *Dm*TMEM63 in the *Tmem63* mutant

background (Fig. 5e). These data demonstrate that loss of *Tmem63* function results in progressive motor impairment with age.

We further tested for the effect of nutritional stress by raising larvae in the starvation condition (Extended Data Fig. 8a). We found that most of the mutant larvae died (Extended Data Fig. 8b) and the remaining viable ones exhibited climbing defects at day 4 of adulthood (Extended Data Fig. 8c), a stage in which well-fed mutant flies performed well (Fig. 5e). Thus, the nutritional stress accelerates the manifestation of the motor impairment in the *Tmem63* mutant flies.

Mutations in the human TMEM63 homologues TMEM63A and TMEM63C are associated with hypo-myelinating leukodystrophy[34] and hereditary spastic paraplegia[35], respectively. The phenotypes of *Tmem63* mutant flies mimic motor disabilities seen in patients with these diseases, prompting us to use this fly animal model to explore the as-yet-unknown pathological mechanisms of TMEM63-related diseases. Since *Dm*TMEM63 is widely expressed in the *Drosophila* nervous system (Extended Data Fig. 2), we examined the effect of *Tmem63* deficiency on the adult neuromuscular junction (NMJ). The pre-synaptic active zone in the motor axon terminals was visualized via immunostaining of the pre-synaptic marker protein Bruchpilot (BRP; Fig. 5f). The synaptic boutons of *Tmem63* mutants appeared normal at day 4 (Fig. 5f,g), but were dramatically reduced by day 30 (Fig. 5f,g), a defect that was rescued by re-expressing the WT *Dm*T-MEM63 in *Tmem63* mutants (Fig. 5f,g). Thus, *Tmem63* mutant flies exhibited progressive synaptic loss, a neuronal defect associated with motor decline in behaviour.

Next, we investigated lysosomal function in the nervous system of *Tmem63* mutant flies. Given that p62, an autophagic adaptor protein

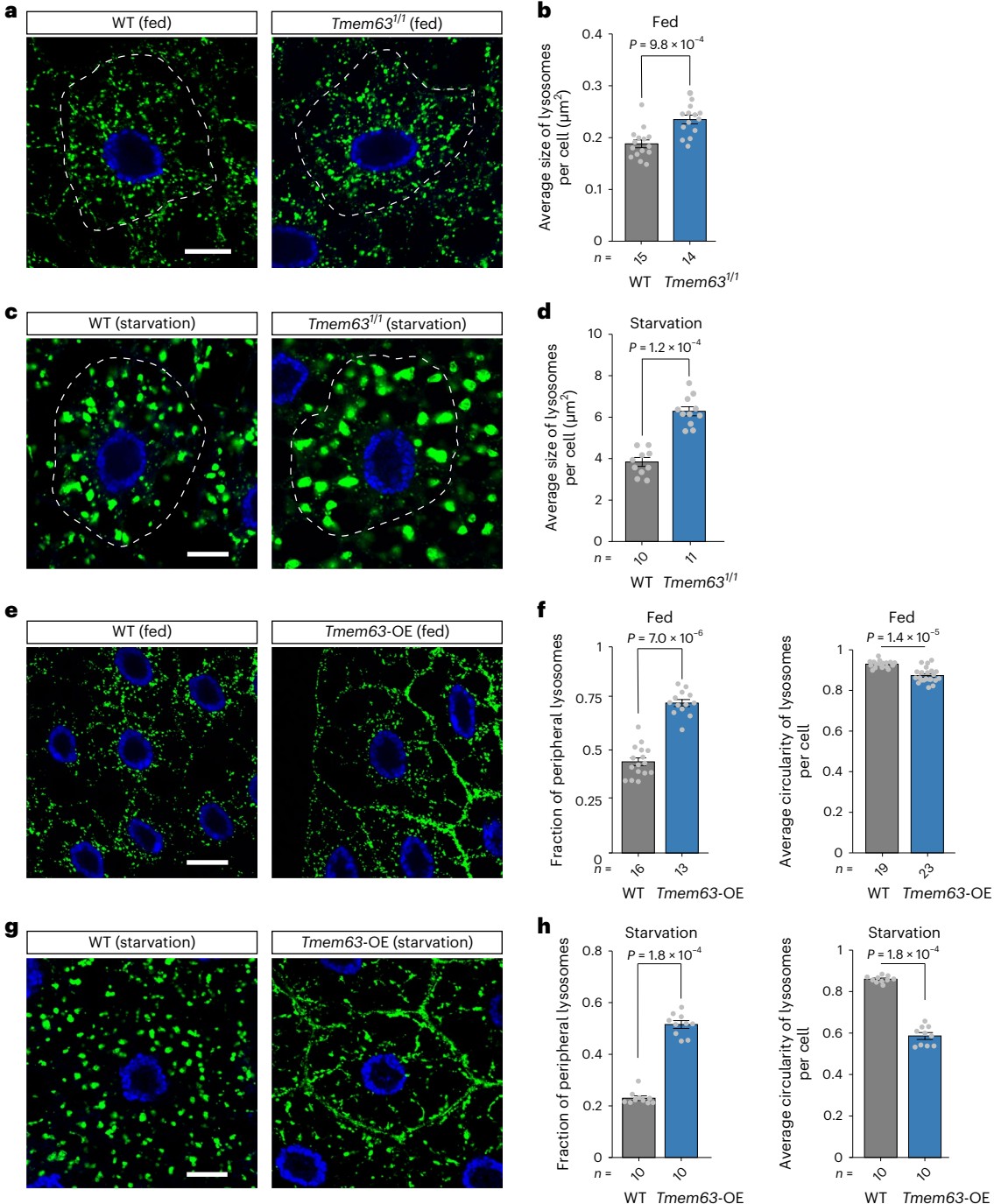

**Fig. 4 | Alternation of *Dm*TMEM63 expression remodels lysosomal morphology. a**–**d**, Images (**a** and **c**) and quantifications (**b** and **d**) of lysosomes (marked by GFP–LAMP1) in fat body cells of the mid L3-stage WT or *Tmem63* mutant larvae under fed or 4 h starved conditions. **e**–**h**, Images (**e** and **g**) and quantifications (**f** and **h**) of lysosomes in mid L3-stage larval fat body cells under fed or 4 h starved conditions. Fat-body specific *Tmem63*-OE was achieved using Cg-Gal4 driver. Lysosomal distribution and lysosomal shape were quantified as the fraction of lysosomes in the periphery and the circularity of lysosomes, respectively. The cell nuclei are labeled in blue. Scale bars, 10 μm (**a**, **c**, **e** and **g**). In **a** and **c**, dashed lines indicate a single cell within the tissues. In **b**,**d**,**f** and **h**, data shown are mean ± s.e.m.; the statistical significance was determined by two-sided Mann–Whitney test; the numbers of cells are shown beneath the bars. The genotypes of the flies are presented in Supplementary Table 1. Numerical data are available as Source data.

that is specifically degraded in lysosomes, is commonly used as a substrate probe to test the capacity of lysosomal degradation[36], we monitored the protein level of Ref2P, the *Drosophila* p62 homologue, in the fly neuronal tissues during aging. At day 4, the Ref2P level in *Tmem63* mutants was nearly twice of that in WT control flies (Fig. 5h,i). Compared with age-matched controls, the Ref2P level in *Tmem63* mutants was 4.5-fold greater at day 16 (Fig. 5h,i) and around 7-fold greater at

day 30 (Fig. 5h,i). This progressive accumulation of Ref2P is indicative of impairment of lysosomal degradation in the *Tmem63* mutant flies.

## Evolutionary conservation of the localization and function of TMEM63 homologues

The *Dm*TMEM63 protein is related to three mammalian TMEM63 proteins (Fig. 6a). In search for a system to examine the functions

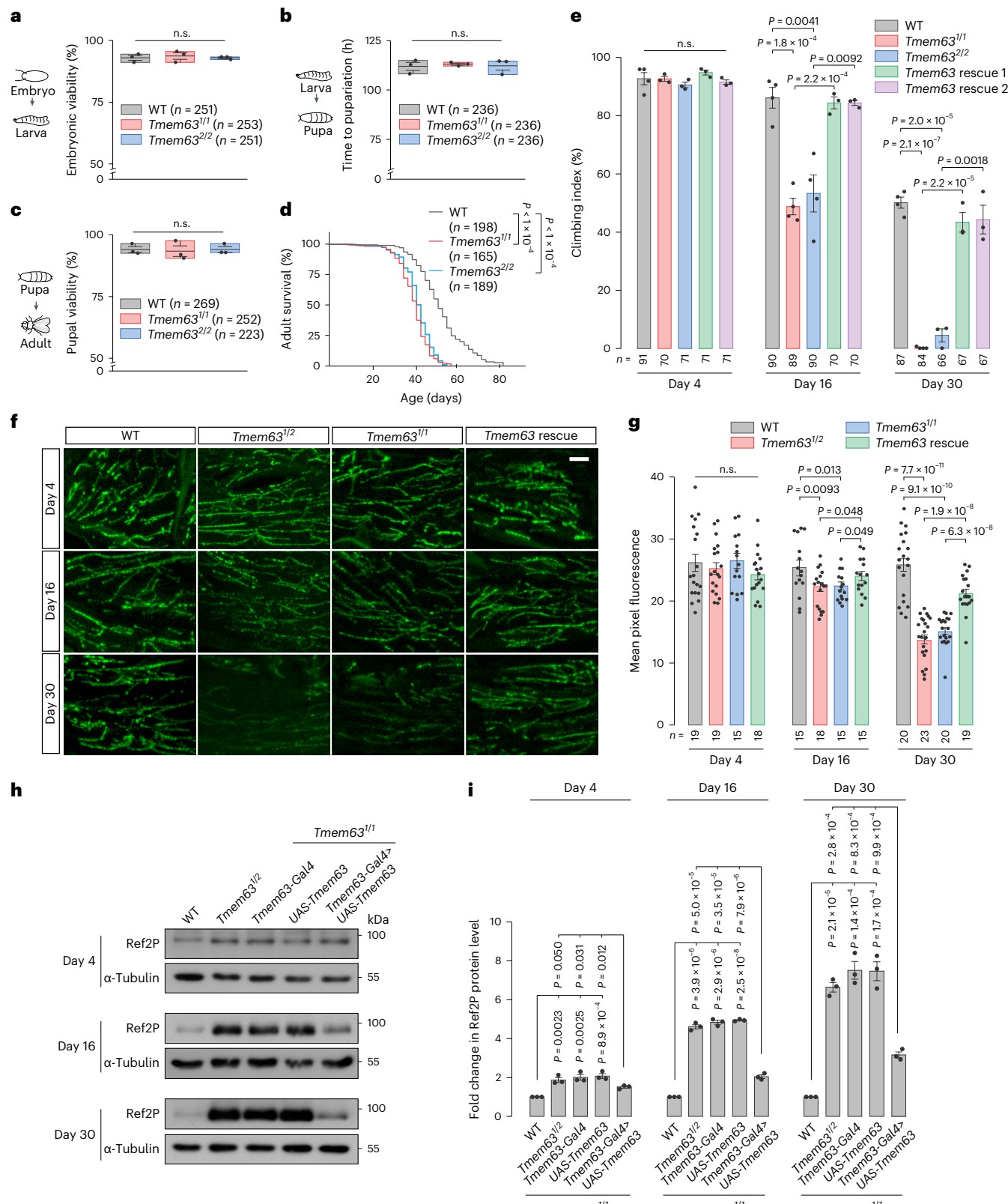

**Fig. 5 | *Tmem63* mutant flies display progressive motor deficits and synaptic loss. a**, Percentage of viable fly embryos. **b**, Developmental time of pupariation. **c**, Percentage of viable pupae. **d**, Lifespan curves of adults. **e**, Climbing activities of adults. **f,g**, Images (**f**) and quantifications (**g**) of adult NMJ. Scale bar, 20 μm. **h,i**, Immunoblots (**h**) and quantifications (**i**) of Ref2P proteins in adult neuronal tissues. *n* = 3 independent experiments. Data shown are mean ± s.e.m. (**e**, **g** and **i**). n.s., not significant. Statistical significance was determined by two-sided log-rank test (**d**), one-way analysis of variance (ANOVA) (**a**–**c** and **e**) or one-way ANOVA adjusted with two-sided Dunn–Šídák test for multiple comparisons (**g** and **i**). In **a**–**c**, boxes edge the interquartile range, lines represent the mean, and whiskers indicate the range of standard error. The numbers of animals (**a**–**e**) or tissues (**g**) are shown. The fly genotypes in **e**–**g** are indicated in Supplementary Table 1. Numerical data and unprocessed blots are available as Source data.

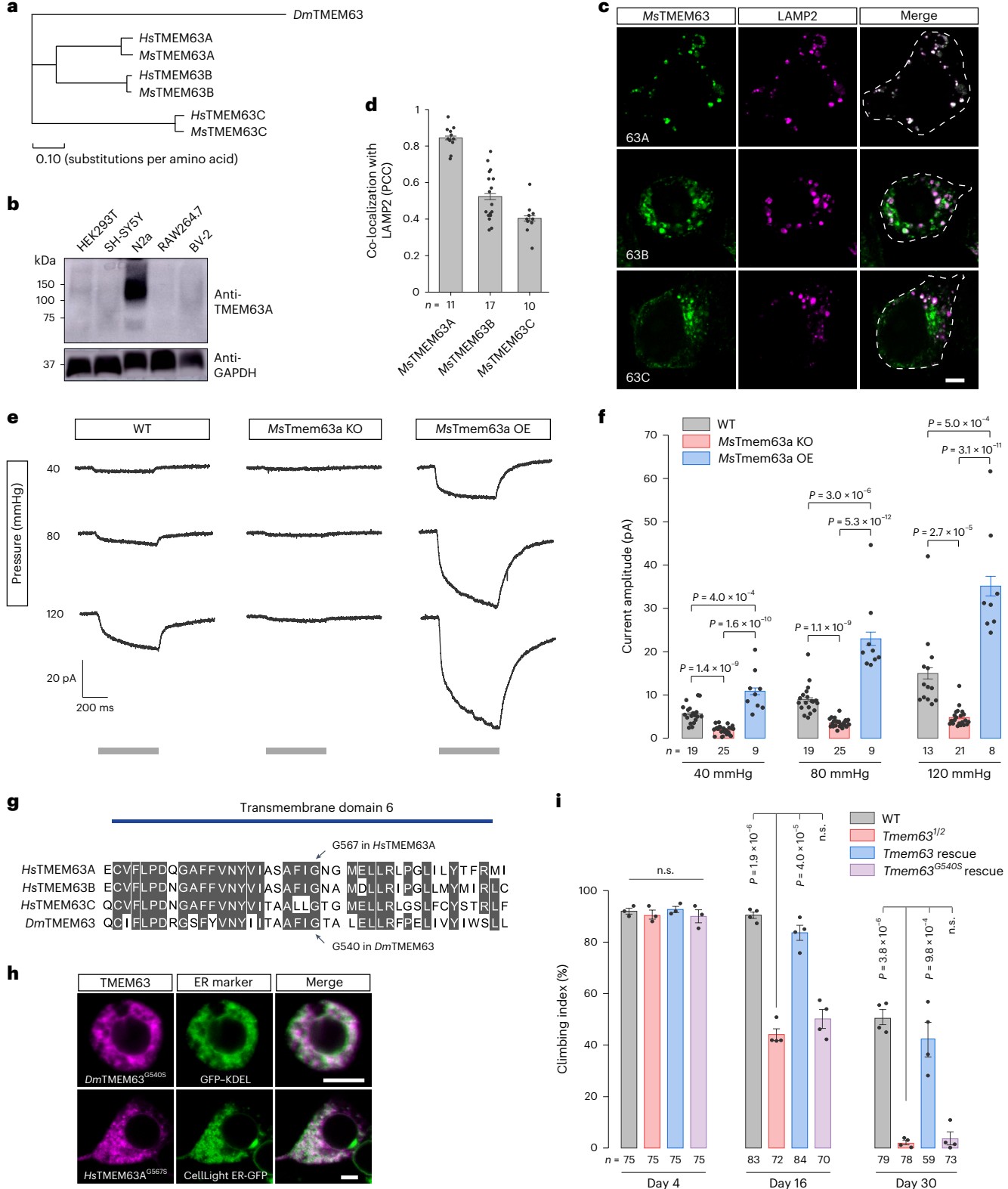

**Fig. 6 | Evolutionary conservation of TMEM63 homologues. a**, Phylogenetic relationship between TMEM63 homologues. Dendrogram was generated using the Phylogeny tool at EMBL-EBI. **b**, Expressions of TMEM63A in mammalian cells. The result is representative of three independent experiments. **c**, Expression patterns of *Ms*TMEM63 proteins in N2a cells with lysosomal marker LAMP2. **d**, Quantification of co-localization of LAMP2 and *Ms*TMEM63 proteins. **e,f**, MS currents (**e**) and quantifications (**f**) of lysosomal recordings. Grey bars indicate time duration of the pressure clamp. **g**, Sequence alignment of the sixth transmembrane domains of TMEM63 proteins. Conserved amino acids are highlighted. Arrows indicate the disease-associated sites. **h**, Expression patterns of TMEM63 mutant proteins in cells with ER markers. **i**, Climbing activities of adult flies. All data shown are mean ± s.e.m. n.s., not significant. Statistical significance was determined by one-way analysis of variance (ANOVA) (**f**) or one-way ANOVA adjusted with two-sided Dunn–Šídák test for multiple comparisons (**i**). The numbers of cells (**d** and **f**) or animals (**i**) are shown. Scale bars, 5 μm (**c** and **h**). The fly genotypes in **i** are indicated in Supplementary Table 1. Numerical data and unprocessed blots are available as Source data.

of mammalian TMEM63 proteins, we screened several cell lines and found Neuro-2a (N2a) cells, a mouse neuroblastoma cell line, showed high-level expression of endogenous *Ms*TMEM63A protein (Fig. 6b). We further examined the localization pattern of *Ms*TMEM63 proteins in N2a cells and found the *Ms*TMEM63 proteins were abundant in the intracellular compartments (Fig. 6c). Of the three mouse TMEM63 proteins, *Ms*TMEM63A showed high co-localization with the lysosomal marker LAMP2 (Fig. 6d). These findings indicate that the lysosomal localization of TMEM63 homologues is conserved between fly and mouse.

To test whether *Ms*TMEM63A is functional in the lysosomal membrane, we performed patch-clamp recordings on the enlarged lysosomes of N2a cells (Extended Data Fig. 9a,b). We detected MS currents from the lysosomal membrane of N2a cells (Fig. 6e). We then knocked out *Ms*Tmem63a gene through clustered regularly interspaced short palindromic repeats (CRISPR)–Cas9-mediated gene deletion in N2a cells (Extended Data Fig. 9c–f) and found that the MS currents were largely abolished in *Ms*Tmem63a-KO cells (Fig. 6e,f). Furthermore, overexpression of *Ms*TMEM63A gave rise to large MS currents in the lysosomal membrane (Fig. 6e,f). These results demonstrate that mammalian lysosomes, just like *Drosophila* lysosomes, are intrinsically MS, and TMEM63A forms a lysosomal MS channel in mammalian cells. Moreover, *Dm*TMEM63$^{G540S}$, bearing a pathogenic mutation of a glycine residue[34] that is conserved between fly and human (Fig. 6g), mislocalized to the endoplasmic reticulum (ER; Fig. 6h) and failed to rescue the motor deficits of *Tmem63* mutant flies (Fig. 6i), suggesting this disease-associated mutation compromises *Dm*TMEM63 protein localization and function. Together, these results suggest that the function of TMEM63 proteins appears to be conserved between fly and mammals.

## Discussion

In recent years, rapid progress has been made in the field of mechanotransduction. While force sensing in the plasma membrane has been intensively studied[12], the mechanisms and physiological roles of intracellular mechanotransduction are just beginning to be elucidated. Our findings reveal a MS channel-mediated mechano-to-electric signal transduction in the lysosomal membrane. Such ultrafast mechanotransduction provides a potential means to detect rapid changes of the local membrane tension during lysosomal remodelling, recycling and trafficking.

What might be the physiological role of *Dm*TMEM63 channel-mediated lysosomal mechanotransduction? Since the Ca$^{2+}$ concentration in the lysosomal lumen is much higher than that in the cytosol[16], one tantalizing hypothesis would be that *Dm*TMEM63 channels mediate Ca$^{2+}$ efflux from lysosomes in response to changes in the mechanical state of the lysosomal membrane. Lysosomal Ca$^{2+}$ release in turn regulates lysosomal remodelling that is essential for the morphological homeostasis and normal functions of lysosomes[15,21,22] (a proposed model is shown in Extended Data Fig. 10). In support of this hypothesis, *Tmem63*-deficient cells displayed enlarged lysosomes, whereas *Tmem63*-OE cells exhibited peripherally positioned, tubular lysosomes. Interestingly, these phenotypic responses are influenced by the nutritional state. Given the fact that lysosomal remodelling and mobility are more dramatic and frequent in the starvation condition than those in the fed condition[37], it is conceivable that lysosomes are subjected to more dynamic forces in the starvation condition. Thus, the nutritional stress seems to boost the role of *Dm*TMEM63 in maintaining lysosomal homeostasis and function, which is evident by the findings that the starved mutant flies showed more severe morphological abnormalities of the lysosomes and earlier manifestation of the behavioural defect.

A positive correlation between the lysosomal curvature and lysosomal Ca$^{2+}$ efflux suggests that *Dm*TMEM63 functions in accordance with lysosomal morphological remodelling. Since the local curvatures

of protrusions or concavities, rather than the global curvature of the whole membrane, control the opening of MS channels[38], it will be interesting in future studies to monitor lysosomal dynamics on a platform equipped with super-resolution microcopy. Previous reports suggest that different types of Ca$^{2+}$-sensitive proteins reside in different subregions of the lysosomal membrane, to determine the region-specific effect of the lysosomal Ca$^{2+}$ release[9,39,40]. It will be desirable to search for the putative Ca$^{2+}$ sensor that is associated with *Dm*TMEM63 to examine the downstream signals of lysosomal mechanotransduction. In addition, changes in the cytoplasmic osmolarity, yielding a physical stimulus involving mechanical force, result in the movement of ions and osmolytes across the membranes of organelles including the lysosome[41,42], yeast ER[43] and the plant plastid[44], suggesting that the osmo- or mechano-transduction may be found in multiple types of organelles.

TMEM63 is a recently discovered ion-channel protein family acting as osmosensors[31,45] or mechanosensors[29,46], with its physiological functions—especially in animals—just beginning to be understood. Flies express only one *Tmem63* gene, greatly facilitating the in vivo study of TMEM63 without confounds of potential functional redundancy of related family members. The broad tissue expression of *Dm*TMEM63 suggests that it may play an important role in general cell physiology besides its specific functions in sensory organs[47,48]. Indeed, we found a high level of the endogenous *Dm*TMEM63 protein in the lysosomal membrane. Although we cannot exclude the possibility that its cell-surface expression is too sparse to be detected, recording from the fat-body plasma membrane revealed no *Dm*TMEM63-mediated MS currents. The demonstration of lysosomal localization of *Dm*TMEM63, and the requirement of *Dm*TMEM63 in lysosomal mechanosensitivity and lysosomal homeostasis, provides evidence that *Dm*TMEM63 is an organellar channel essential for lysosomal morphology and function.

So far, most symptoms found in MS channel-related diseases concern the pathology in sensory transduction[11]; the association of a MS channel with motor disorders has been rarely reported and the pathogenesis is largely unknown. In this study, we found that a pathogenic mutation of the TMEM63 family caused the mutant proteins to be retained in the ER, suggesting a misfolding or trafficking-defective phenotype of this mutant protein. Moreover, the phenotypes found in *Tmem63* mutant flies mimic some aspects of the symptoms in TMEM63-associated diseases. It is important to note that, of the phenotypes observed in *Tmem63* mutants, impaired lysosomal degradation precedes the occurrences of synaptic loss and motor defect, suggesting an important role of lysosomal dysfunction in the initial stage of TMEM63-related diseases. Similarly, malfunction of lysosomal degradation was reported as an early pathological event to induce amyloid plaques in Alzheimer's disease[49]. Targeting lysosomal proteins can ameliorate neuropathology in animal models of lysosomal storage diseases and neurodegeneration[50,51]. It is of potential therapeutic significance to determine the lysosomal pathology in mammalian cells with TMEM63 mutations (for example, patient induced pluripotent stem cell-derived neurons) and evaluate the effects of lysosome-targeting chemicals on these cells. In summary, identifying a lysosomal mechanosensor advances the fundamental understanding of lysosomal physiology and provides insight into the interplay between intracellular mechanotransduction, organellar dynamics and organelle-related pathologies.

## Online content

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

## Methods

### Molecular cloning

To clone the MS channels, *DmPiezo* was amplified from the genomic DNA of the UAS−*Piezo*−GFP transgenic fly. The *nompC* complementary DNA (cDNA) was cloned in the lab before. The *Iav* cDNA was generated through gBlocks gene synthesis (IDT). Each gene was subsequently inserted into pUAST−mCherry vector using Gibson assembly kit (NEB). The *Tmem63* cDNA was amplified from *Drosophila* Genomic Resource Center (DGRC, stock no. 9185) and used to generate pUAST−*Tmem63*−mCherry, pJFRC81−*Tmem63*−GFP and pAC−*Tmem63*−mCherry−GCaMP6f plasmids. *DmTmem63*[G540S] was generated by site-directed mutagenesis using Phusion DNA polymerase (NEB) and verified by DNA sequencing of the entire coding region.

The plasmids containing Mito−YFP or GFP−SKL were from Vladimir Gelfand lab (Northwestern University). The plasmid pUASp−Lys−GFP−KDEL was from Mary Lilly lab (National Institutes of Health (NIH)/National Institute of Child Health and Human Development (NICHD). The constructs LAMP2−mCherry and LAMP1−YFP were from Li Yu lab (Tsinghua University). *Drosophila Lamp1* cDNA was amplified from the construct UAS−LAMP1−RFP (Chao Tong lab, Zhejiang University) and used to generate pAC−*lamp1*−mCherry−GCaMP6f and pJFRC81−*lamp1*−GFP construct, respectively.

Mammalian TMEM63 clones were purchased from ORIGENE and GenScript: *Ms*Tmem63a (NM_144794, cat no. MR210748), *Ms*Tmem63b (NM_198167, cat no. MR221527), *Ms*Tmem63c (NM_172583, cat no. MR210738) and *Hs*Tmem63a (NM_014698, cat no. OHu08349). The cDNAs were then subcloned into pmCherry−N1, pEGFP−N1 or pSIN−EF2−3xFlag−puro vector. The point mutation in *Hs*Tmem63a[G567S] was introduced by site-directed mutagenesis using Phusion DNA polymerase (NEB).

### Fly stocks and husbandry

The following fly strains were obtained from the Bloomington Stock Center: stock no. 39668 (UAS−*spin.myc*−GFP), stock no. 7011 (Cg-GAL4), stock no. 48183 (GMR51B08-GAL4), stock no. 32194 (UAS−mCD8−GFP) and stock no. 58772 (UAS−*Piezo*−GFP). The stock UAS−GFP−LAMP1 (ref. 52) was from Helmut Kramer lab (UT Southwestern). The stock Cg-GAL4, UAS−GFP−LAMP1 (ref. 53) was from Amy Kiger lab University of California San Diego(UCSD). The control fly stock was *w*[1118]. All the mutant fly strains were backcrossed with *w*[1118] background for at least five generations. The well-fed flies were raised on standard cornmeal−molasses food consisted of (per 750 ml) 8.1 g agar, 11.225 g yeast, 45 g cornmeal, 60 ml molasses, 6 ml tegosept, 2.15 ml propionic acid and 0.425 ml phosphoric acid. The starved adult flies were raised on low-nutrient food consisted of (per 750 ml) 25.6 g sucrose, 8.1 g agar, 6 ml tegosept and 2.15 ml propionic acid.

### Generation of gene-modified flies

The *Tmem63*-Gal4 and UAS−*Tmem63*−GFP transgenic flies were generated through P-element-mediated germline transformation. The UAS−*Tmem63* transgenic fly was generated by phiC31-mediated chromosomal integration. The fly injections were performed by Genetivision.

The *Tmem63*[1] mutant was generated via the CRISPR−Cas9 system. To assemble the KO donor construct, the 5′ and 3′ homologous arms were amplified from *w*[1118] genomic DNA and inserted into the pHR−EGFP vector using Gibson assembly kit (NEB). The two guide RNAs were cloned into the pU6−BbsI−chiRNA vector. A mixture of guide RNAs and donor plasmids were injected into vas−cas9 flies (Rainbow Transgenics). F1 flies with red eyes were selected and crossed to balancers. The *Tmem63*[2] mutant was generated through ends-out homologous recombination[54]. The 5′ and 3′ homologous arms were amplified from *w*[1118] genome and inserted into the pw35Gal4 vector[55]. The donor construct was transformed into the germline of *w*[1118] flies through P-element-mediated germline transformation. Transgene on

the third chromosome was selected for further cross and heat-shock induced excision. The *Tmem63*[2] allele with red eyes was then isolated.

We used the CRISPR−Cas9 system to insert GFP or mCherry in place of the stop codon of *Tmem63* gene in its native locus. The two homology arms were amplified from the genomic DNA of *w*[1118] and assembled into the donor vector pHR−EGFP or pHR−mCherry using Gibson assembly kit (NEB). The guide RNA was synthesized and inserted into pU6−BbsI−chiRNA vector. The guide RNA and donor plasmid were co-injected into vas−cas9 flies (Rainbow Transgenics). Knock-in lines were verified by genotyping and western blotting. The oligos used in generating the mutant and knock-in flies are listed in Supplementary Table 2.

### Cell culture and generation of stable cell lines

S2 cells (ATCC, no. CRL-1963) were cultured in Schneider's *Drosophila* Medium (ThermoFisher) supplied with 10% foetal bovine serum (Gibco) and 1× penicillin−streptomycin (ThermoFisher) in 25 °C incubator. N2a cells (ATCC, no. CCL-131), HEK293 cells (ATCC, no. CRL-1573), HEK293T cells (ATCC, no. CRL-3216), SH-SY5Y cells (ATCC, no. CRL-2266) and BV-2 cells (ICLC, no. ATL03001) were cultured in Dulbecco's modified Eagle medium (Gibco) supplemented with 10% foetal bovine serum and penicillin−streptomycin at 37 °C in 5% $CO_2$ incubator. Cells were plated into 3.5-cm Petri dishes or six-well plates at an approximately 70% cell density before transfection using jetOPTIMUS DNA Transfection Reagent (Polyplus).

Lentivirus transfection was used to generate stable cell lines as described before[56]. The lentiviral particles were produced by co-transfection of the expression vector (pSIN−EF2−*Ms*Tmem63a−puro) or gene-deletion vector (pLenti−CRISPR−*Ms*Tmem63a-targeting sequence) with the packaging construct (psPAX2) and envelope construct (pMD2.G) into HEK293 cells. At day 2 after transfection, the culture medium containing the lentiviral particles were collected, filtered and added to the N2a cells cultured in the antibiotic-free culture medium with polybrene (8 μg ml⁻¹; Sigma-Aldrich). The infected cells were selected with puromycin (3 μg ml⁻¹) for three passages, and the single-cell clones were sorted and expanded to establish stable cell lines. The *Ms*TMEM63A-expressing stable line was further transiently transfected with *Ms*TMEM63A−GFP construct to achieve overexpression of *Ms*TMEM63A in N2a cells for patch-clamp recordings (see below). The CRISPR−Cas9-targeted exon sequence in *Ms*Tmem63a was GGATTATGGCCGCATAGCCC[TGG] (with TGG serving as the protospacer adjacent motif).

### Electrophysiology

For cell-attached recordings on S2 cells, the borosilicate glass pipettes (Sutter instrument, BF120-69-10) were used with resistances in a range of 3–4 MΩ. To monitor the dose-dependent response curve of the MS currents, a 500 ms stepwise protocol with 10 mmHg pressure increment was applied to the cell membrane through the recording pipette with a high-speed pressure clamp (HSPC, ALA-scientific). For cell-attached recordings on the fat-body plasma membrane, the fat bodies were dissected from the late-L3 stage larvae and placed on the poly-L-lysine-coated coverslips. A stretch (from 0 to −95 mmHg with 5 mmHg increment) was applied on the cell membrane. The solutions used in these cell-attached recordings are as follows: base solution containing (in mM) 140 KCl, 1 $MgCl_2$, 10 glucose and 10 HEPES (pH 7.4 with KOH) and the standard pipette solution containing (in mM) 133 NaCl, 5 KCl, 1 $CaCl_2$, 1 $MgCl_2$ and 10 HEPES (pH 7.4 with NaOH). For ion substitution experiments in S2 cells, the NaCl pipette solution was (in mM) 150 NaCl and 10 HEPES (pH 7.4), the KCl pipette solution was (in mM) 150 KCl and 10 HEPES (pH 7.4), and the NMDG-Cl pipette solution was (in mM) 150 NMDG and 10 HEPES (pH 7.4). The osmolarity of all the solutions was adjusted to 300 ± 3 mOsm kg⁻¹.

The method to perform recordings on single native lysosome of the fat body was modified from enlarged endolysosomal recording[57,58] and single mitoplast recording[59]. Fat bodies were dissected from the

late-L3 stage larvae and cut into small pieces by spring scissors (Fine Science Tools, 15000-00). The lysosomes were released from the cytosol to the bath solution in the recording chamber. After lysosomes settled on the poly-L-lysine-coated coverslip (~20 min), fresh bath solution was perfused to remove tissue debris floating on the surface of the solution. Recordings were performed under Olympus microscope (BX51W1) equipped with water-immersion lens and video cameras. The intact lysosomes of WT fat bodies were spherical with diameters around 3 μm, which is accessible for patch-clamp recordings, whereas the lysosomes in fat bodies overexpressing *Tmem63* were in irregular shapes and could not form gigaseals with the recording pipettes. The lysosome-attached patch-clamp recordings were performed using borosilicate glass recording pipettes (Sutter Instrument BF150-75-10) with resistances in the range of 6–8 MΩ. The base solution used to balance the lysosomal membrane potential contains (in mM) 120 NaCl, 20 KCl, 0.5 CaCl$_2$ and 10 glucose (pH 7.2 with NaOH). The solution in the recording pipette contains (in mM) 140 K-gluconate, 4 NaCl, 2 MgCl$_2$, 1 egtazic acid, 0.39 CaCl$_2$ and 10 HEPES (pH 7.2 with KOH).

Recordings on the enlarged lysosomes were performed according to standard methods[57,58] with slight modifications. Consistent with a previous report[20], the S2 cell lysosomes could not be enlarged to a size that is suitable for patch-clamp recordings. However, the lysosomes in N2a cells could be enlarged through a combination of vacuolin-1 treatment (1 μM, Millipore Sigma) and transfection of a constitutively active form of Rab5 (*Hs*Rab5$^{Q79L}$). Enlarged lysosomes were then dissected out of the cells and visualized under the microscope (IX73, Olympus) before the lysosome-attached patch-clamp recordings. Given that the enlarged lysosomes were too fragile to endure repeated mechanical stretch, a three-step protocol (from −40 to −120 mmHg with 40 mm Hg increment) was applied in the recordings. The base solution and pipette solution were the same as those used in the lysosomal recordings of fat-body lysosomes.

All the electrophysiological recordings were performed at room temperature with an Axopatch 700B amplifier and a Digidata 1440A or Digidata 1550B digitizer (Molecular Devices). The currents were sampled at 10 kHz and filtered at 0.5 kHz (low pass). PClamp10.4, Clampfit 10.4 (Molecular Devices) and Origin 2021 (OriginLab) softwares were used to acquire and analyse data. The single-channel conductance was estimated through the non-stationary noise analysis as described previously[60].

### Antibodies
The primary antibodies used in the immunostaining experiments were as follows: mouse anti-GFP antibody (1:500, Roche, no. 11814460001), chicken anti-mCherry antibody (1:300, Novus Biologicals, no. NBP2-25158) and mouse anti-BRP antibody (1:100, Developmental Studies Hybridoma Bank, no. nc82). The secondary antibodies for immunostaining were anti-mouse labelled by Alexa 488 (1:1,000, ThermoFisher no. A28175) or Cy3 (1:1,000, Jackson ImmunoResearch Labs no. 115-165-146) and anti-chicken labelled by Alexa 647 (1:1,000, Jackson ImmunoResearch Labs, no. 103-605-155). The primary antibodies used in the western blots were mouse anti-GFP antibody (1:2,000, Roche, no. 11814460001), rabbit anti-mCherry antibody (1:2,000, Abcam, no. ab167453), rabbit anti-Ref2P antibody (1:500, Abcam, no. ab178440), rabbit anti-TMEM63A antibody (1:200, Novus Biologicals, no. NBP2-57359), mouse anti-GAPDH antibody (1:2,000, Proteintech, no. 60004-1-Ig) and mouse anti-tubulin antibody (1:2,000, Sigma, no. T9026). The secondary antibodies for the western blots were anti-mouse HRP antibody (1:4,000, Jackson ImmunoResearch Labs, no. 115-035-146) and anti-rabbit HRP antibody (1:4,000, Jackson ImmunoResearch Labs, no. 111-035-144).

### Immunostaining and confocal imaging
The fly tissue staining was performed as described previously[61] with slight modifications. Briefly, fly tissues were dissected in cold phosphate-buffered saline (PBS) and fixed in 4% formaldehyde (Sigma-Aldrich) at room temperature for 20 min. For direct observation of the fluorescent signals (Fig. 4 and Extended Data Fig. 2a), fly tissues were washed three times in PBS. The tissues were then mounted in SlowFade Mountant (ThermoFisher) for confocal imaging. For staining with antibodies (Fig. 1e,f), samples were blocked in blocking buffer (0.3% Triton X-100 and 10% normal goat serum in PBS) for 30 min at room temperature and then incubated with primary antibody overnight at 4 °C. After three washes in PBST (0.3% Triton X-100 in PBS), samples were incubated in secondary antibodies for 2 h at room temperature. After three washes in PBST and one wash in PBS, tissues were mounted for imaging.

For NMJ staining, thoraxes of the adult flies were processed by cryostat before stained with antibodies. The adult flies were first washed in PBS, 100% ethanol (KOPTEC, no. V1016) and PBS, and then thoraxes were dissected in PBS and fixed in PBST with 4% formaldehyde at room temperature for 10 min. The tissues were placed subsequently in 15% sucrose (Fisher Scientific, no. S5-3,) until they sank and then 30% sucrose until they sank for cryopreservation. The tissues were placed and embedded in O.C.T. Compound (Fisher Scientific, no. 23-730-571) before a cryo-treatment in dry ice sprayed with ethanol. Thoraxes were sliced longitudinally into 50-μm pieces by cryostat (Leica, CM 3050S).

For imaging cultured cells, cells were grown on coverslips coated with poly-L-lysine. Forty hours (for S2 cells) or 30 h (for N2a and HEK293 cells) after transfection with plasmids, cells were fixed with 4% paraformaldehyde for 20 min at room temperature. After three washes with PBS, cells were mounted onto microscope slides for confocal imaging.

The images shown in the Fig. 4 were acquired with an Olympus FV3000 confocal microscope with the FV31S-SW 2.6 software. The other confocal images were captured by a Leica SP8 confocal microscope with the LAS X 4.1.1 software. Greyscale and RGB images were further processed with ImageJ Fiji 2.1.0 (NIH). Pearson's correlation coefficient was measured by Coloc2 plugin of the ImageJ to quantify co-localization index. The Analyze Particles function in ImageJ was used to measure lysosomal size and circularity. The lysosomal distribution was quantified according to previous protocol[62]. The outline of a fat body cell was degraded inward by 5 μm to create two concentric shells. The lysosomes in the overlapping area of the two shells were defined as peripheral lysosomes of the cell.

### Organelle staining
Fly tissues were dissected in cold PBS and transferred to 1 μM LysoTracker Red DND-99 (ThermoFisher, no. L7528) for 3 min at room temperature. For direct imaging without co-staining of antibodies (Extended Data Fig. 7), samples were mounted and imaged within 30 min after dissection. For LysoTracker staining combined with immunohistochemistry (Fig. 1f), the samples were fixed in 4% formaldehyde for 3 min at room temperature and then stained with antibodies. LysoTracker signals in fat body tissues maintained well after co-staining with antibodies, whereas LysoTracker signals in neurons were not observed after immunostaining possibly due to the low penetration and (or) diminishment of the dye. In addition, for LysoTracker staining in the fat bodies of L3-stage larvae, LysoTracker dye could only label the acidic lysosomes that was induced under starvation condition, but not the lysosomes of the fed larvae[32,33].

CellLight ER-GFP (ThermoFisher, no. C10590) was used to label ER in HEK293 cells. CellLight ER-GFP was dissolved in the culture medium (1:50) and incubated with cells overnight. Cells were fixed with 4% paraformaldehyde (Sigma-Aldrich) for 20 min at room temperature. After three washes with PBS, cells were mounted onto the microscope slides for imaging.

### Calcium imaging
S2 cells were plated on Concanavalin A-coated coverslips, incubated in the isotonic solution and imaged at 400 ms per frame under ORCA-Flash4.0 Hamamatsu digital camera C13440 paired with

Olympus IX73 microscope and 60× objective. The Micro-manager 2.0.0 software was used to acquire imaging data. After imaging for 1 min in the isotonic condition, cells were perfused with the hypotonic solution and then imaged for another 9 min. The calcium-free hypo-osmotic solution contained (in mM) 60 NaCl, 5 KCl, 1 MgCl$_2$ and 10 HEPES, pH 7.4 (160 mOsm l$^{-1}$). The isotonic solution (299 mOsm l$^{-1}$) was adjusted with mannitol and contained the same concentration of ions as the hypotonic solution. The calcium fluorescence images were analysed after subtraction of background in ImageJ Fiji 2.1.0. Changes of the fluorescence were shown as the ratio of a real-time intensity relative to the value at the first frame.

### Western blotting

A total of 10–20 fly heads and VNCs were homogenized by tissue homogenizers with pestles (Kimble) in RIPA lysis buffer (50 mM Tris–HCl pH 8.0, 150 mM NaCl, 0.5% Na$_3$VO$_4$, 1% Triton X-100, 0.1% sodium dodecyl sulfate, 1 mM ethylenediaminetetraacetic acid, 1 mM phenylmethylsulfonyl fluoride and protease inhibitor (Roche, no. 11697498001)). Samples were then centrifuged at 4 °C for 10 min at 15,000$g$. Pellets were discarded and the supernatants were mixed with NuPAGE LDS sample buffer (ThermoFisher, no. NP0007) with 2.5% β-mercaptoethanol. After heating at 95 °C for 10 min, protein samples were separated on 4–12% Bis-Tris Plus Gels (ThermoFisher, no. NP0322), transferred to polyvinylidene fluoride membrane, blocked for 1 h at room temperature in blocking buffer (5% milk in Tris-buffered saline with 0.1% Tween-20) and incubated with primary antibodies at 4 °C overnight. After washing with TBST three times, membranes were further incubated with secondary antibodies for 1.5 h at room temperature. Membranes were washed with TBST before enhanced chemiluminescence detection (ThermoFisher, no. 34076). The enhanced chemiluminescence signals were captured by the Odyssey CLx digital imager or Fuji film.

### Starvation treatment

Short-term starvation treatment was performed as described in the previous study[63]. Briefly, larvae were raised at low density (25 larvae per vial). The mid-L3 stage larvae (90 h after egg laying) were either kept in food (fed condition) or acutely starved for 4 h in 20% sucrose in PBS before dissection. For long-term starvation, the mid-L3 stage larvae were picked into an empty vial with a wet filter on the bottom (25 larvae per vial). The viable adults developed from the starved larvae were raised on low-nutritional food before behavioural analyses.

### Development and lifespan assays

For assessing embryonic viability, embryos were collected after 4 h egg laying. The number of hatched embryos (first-instar larvae) and dead embryos was scored 36 h later. Embryonic viability was calculated as the number of hatched embryos divided by the number of total embryos. For assessing larval development, synchronized first-instar larvae (within 4 h of hatching) were picked into vials. The number of the pupae was scored once in the morning and once in the evening. Pupal stage was assessed on the basis of the maturity of the mouth hook. Time to pupariation was calculated as an averaged time for each vial. Pupal viability was calculated as the number of the adult flies divided by the number of pupae. For assessing lifespan, adult flies were collected and transferred to yeast-added vials (20 flies per vial). Flies were transferred to fresh vials every 3–4 days, and the number of the dead flies was recorded. All assays were performed at 25 °C.

### Climbing assay

Age-matched male flies were transferred to empty 15-ml centrifuge tubes (Corning) and acclimatized for 5 min. For each trial, flies were tapped down, and the videos of the climbing behaviour were recorded. The climbing index was defined as the percentage of the flies that were able to cross an 8-cm marker within 10 s. Three trials were recorded for each cohort with a 1-min interval between each trial.

### Analysis of correlation between lysosomal curvature and calcium intensity

S2 cells expressing *Dm*TMEM63–mCherry–GCaMP6f were imaged at 2 Hz under Olympus SpinSR spinning disk confocal microscope with cellSens 4.2 software and 60× objective. A region of interest that includes a discrete whole lysosome was cropped out. The lysosome morphology (mCherry signal) of the cropped region of interest was then imported into Ilastik[64] for segmentation of individual lysosome over time. A custom script in MATLAB R2020a was used to convert the segmentation results into binary masks. A lysosome with the mCherry signal intensity over 300 and object size over 200 pixels was used for further calculation.

A custom MATLAB script was then used to quantify the local curvature and calcium intensity. The pixels on lysosome boundary were extracted from binary masks and used as sampling points to calculate local curvature and calcium intensity. The code for local curvature measurement of each sampling point was modified on the basis of the curvature measure and visualization function[65]. Only sampling points with non-negative curvature were used for further analysis since these regions represent bulged areas on a lysosome. To quantify the local calcium intensity in the proximity of a sampling point, a 5 × 5 pixel (0.11 μm$^2$) local region centred on the sampling point was drawn, and mean intensities of the pixels on the local region and inner side of the boundary were calculated. The normalized calcium intensity was defined as the ratio between the mean intensity of the local calcium (GCaMP6f signal) and the mean intensity of the local lysosome (mCherry signal). A correlation coefficient between curvature and normalized local calcium intensity from all sampling points was calculated as the value for one lysosome at one time point. Correlation coefficients from the same lysosome at all timepoints were then averaged, which provides a frame-averaged correlation coefficient value associated with an individual lysosome.

### Statistics and reproducibility

All experimental data are shown as percentile distribution or mean ± standard error of the mean (s.e.m.), as indicated in the corresponding figure legends. All experiments were performed at least three times with similar results. Statistical analyses were carried out in Origin 2021. Statistical tests, sample sizes and exact $P$ values are provided in the corresponding figures or figure legends. No statistical method was used to pre-determine the sample size. No data were excluded except for the climbing assays in which dead flies were not counted as total numbers of the tested flies. Animals or cells were allocated on the basis of genotypes or treatments, as described in the corresponding figure legends. Images were obtained randomly and were taken from random regions of prepared samples. The experimenters who collected and analysed the imaging and behavioural data were blinded to the group allocations.

### Reporting summary

Further information on research design is available in the Nature Portfolio Reporting Summary linked to this article.

## Data availability

All data of this study are available in the main text or the extended materials with numeric data of graphs and uncropped scans of all blots and gels shown in Source data files. The genotypes of the animals used in figures and the sequences of oligonucleotides used in Methods are listed in Supplementary Tables 1 and 2, respectively. For Extended Data Fig. 2b, datasets analysed are available in the Fly Cell Atlas repository[66,67] (https://scope.aertslab.org/#/FlyCellAtlas/). Source data are provided with this paper.

## Code availability

Two custom MATLAB scripts were used to convert the segmentation results to binary masks and quantify the local curvature and Ca$^{2+}$

intensity. These codes are available at GitHub (https://github.com/weichensteven/lysosome_curvature_calcium_correlation_analysis).

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

## Acknowledgements

We thank A. Kao, C. O'Brien and P. Jin for critical reading of the manuscript and members of the Jan lab for discussion. We thank Z. Liu for help in the noise analysis. We thank C. Ma and L. Guo in the Yichang Jia lab for giving advice on the protein and DNA electrophoresis. We thank V. Gelfand, M. Lilly, L. Yu and C. Tong for gifting plasmids. We thank the Nevan Krogan lab and the Li Gan lab for providing us with N2a cells and BV-2 cells, respectively. We thank the Bloomington *Drosophila* Stock Center for providing us fly stocks. This work was supported by NIH National Institutes of Neurological Disorders and Stroke grants to Y.N.J. (R35NS097227) and L.Y.J. (R35NS122110) and a Human Frontier Science Program fellowship (LT000370/2019-L) to K.L. L.Y.J. and Y.N.J. are investigators of the Howard Hughes Medical Institute.

## Author contributions

Y.N.J. and L.Y.J. supervised the project. W.Z. hosted K.L. during the remote working in the pandemic. Y.G. conceived the project. K.L. and Y.G. generated genetically modified animals. K.L. performed lysosomal recordings, cell-attached recordings, immunostainings, immunoblots and behavioural assays. Y.G. performed recordings on fat-body plasma membrane. Y.W. generated gene-modified cell lines with assistance of Y.G. and K.L. Y.W. performed western blots, immunocytochemistry and lysosomal recordings in N2a cells. Y.W. performed noise analysis with help of X.Z. R.Z. performed NMJ analysis and $Ca^{2+}$ imaging experiments. W.C. performed lysosomal live-imaging experiments with assistance of K.L. T.C. generated some of the constructs. X.Z. assisted in data analysis. Y.J. assisted in imaging experiments. T.L. assisted in behavioural experiments. K.L. wrote the manuscript.

## Competing interests

The authors declare no competing interests.

## Additional information

**Extended data** is available for this paper at https://doi.org/10.1038/s41556-024-01353-7.

**Correspondence and requests for materials** should be addressed to Yuh Nung Jan.

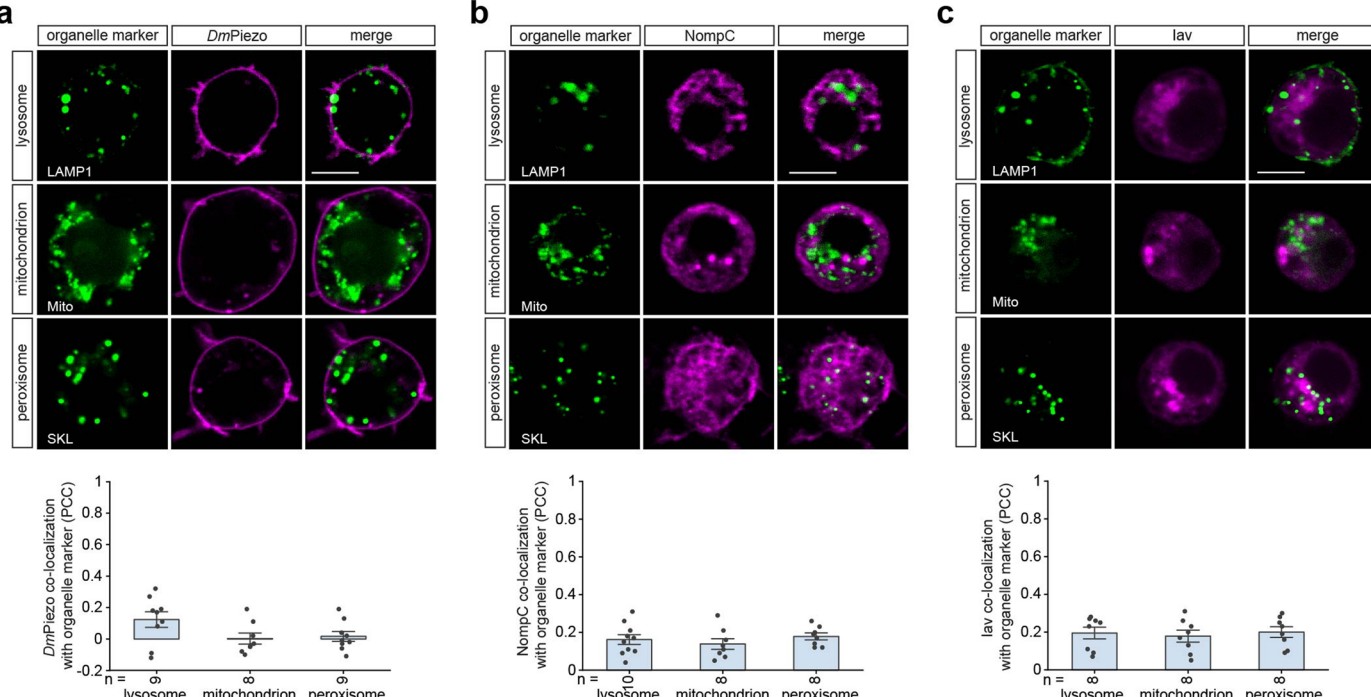

**Extended Data Fig. 1 | Subcellular localization patterns of mechanosensitive channel *Dm*Piezo, NompC, and Iav.** Representative images showing the expression patterns of *Dm*Piezo (**a**), NompC (**b**), Iav (**c**) proteins and organellar markers. Each mCherry-tagged channel protein was co-expressed with LAMP1-GFP, Mito-YFP or GFP-SKL in S2 cells. The co-localization between the channel protein and organellar marker was quantified by Pearson's correlation coefficient (PCC). Data shown are mean ± s.e.m. The numbers of cells are shown beneath the bars. Scale bar, 5 µm. Numerical data is available as source data.

**a**

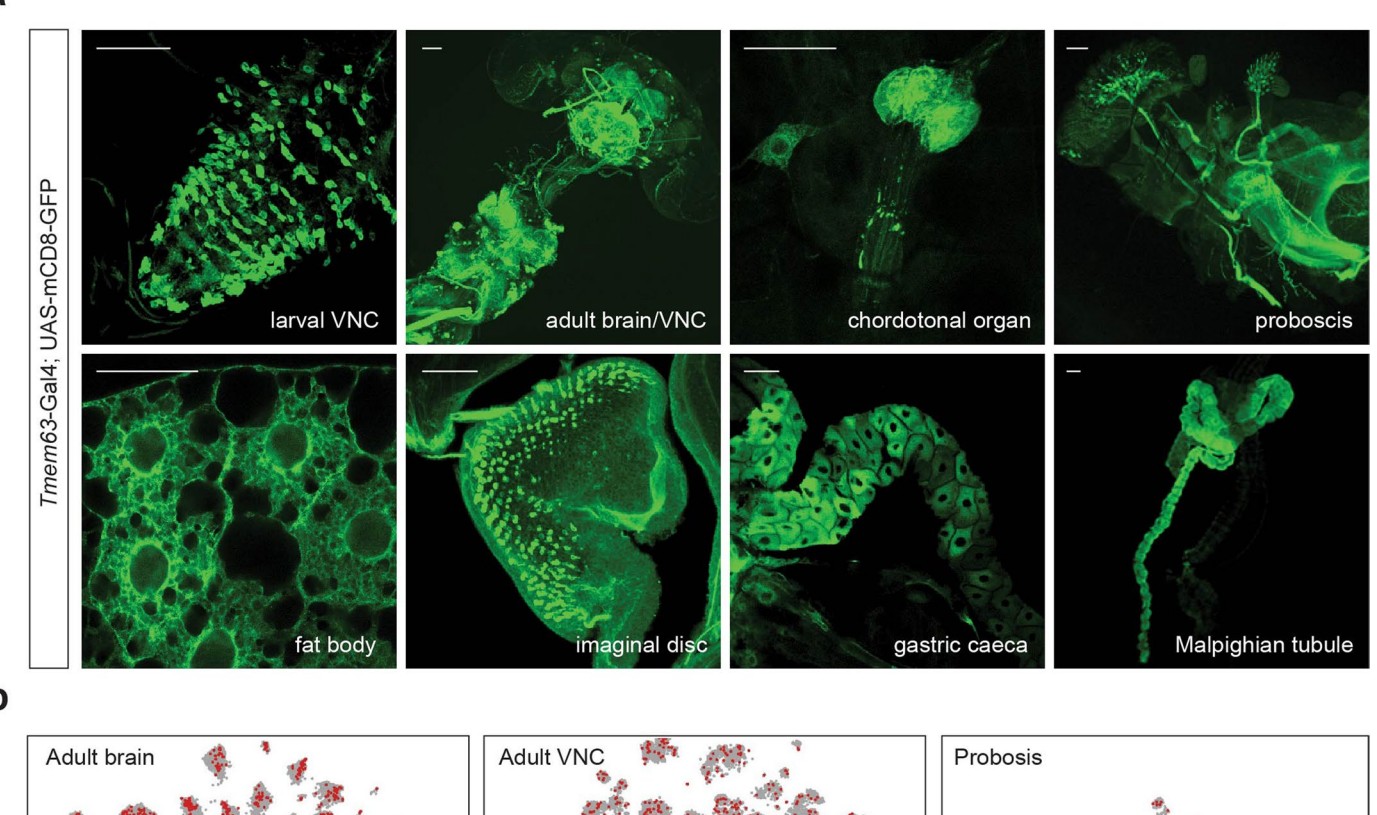

**b**

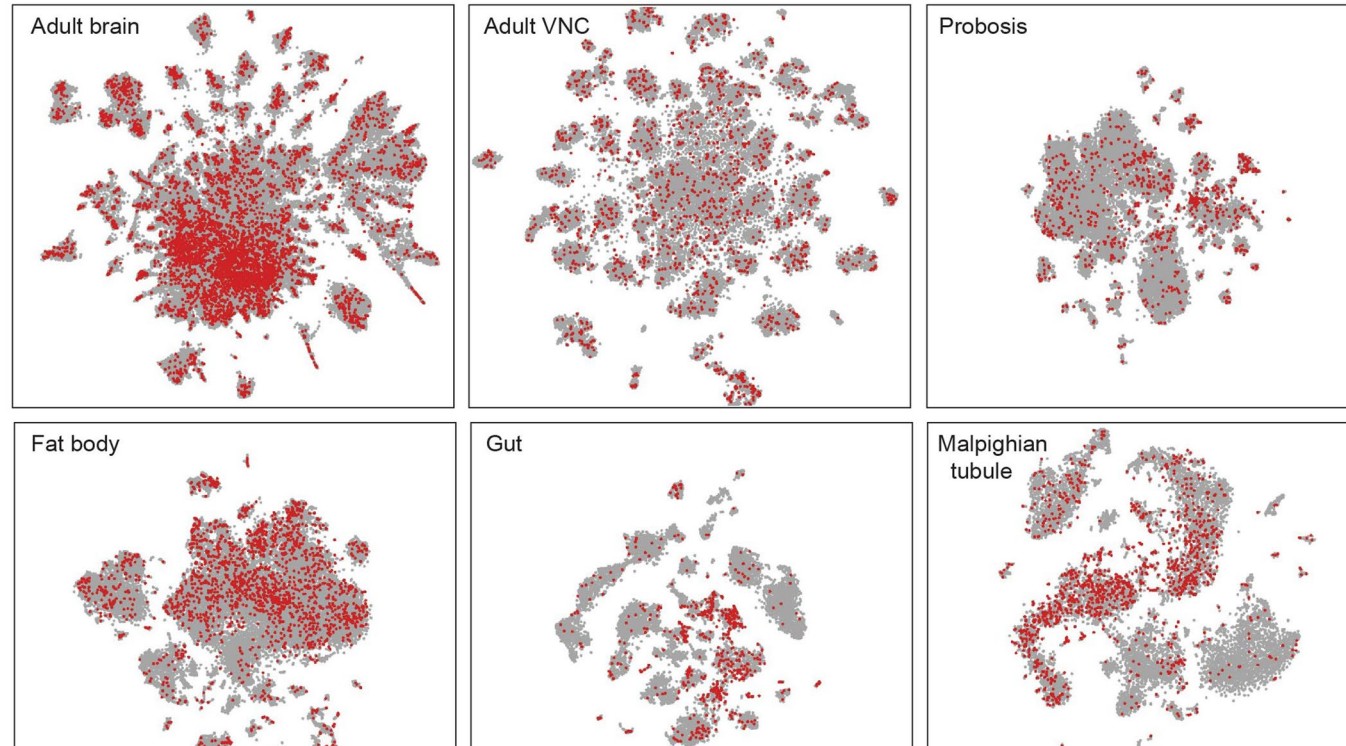

**Extended Data Fig. 2 | *Drosophila Tmem63* is broadly expressed in many tissues. a**, Tissue expression patterns of *Tmem63*-Gal4 reporter (UAS-mCD8-GFP driven by *Tmem63*-Gal4). The images are representatives of three independent experiments. Scale bar, 50 μm. **b**, Transcriptional profile of the *Tmem63* gene. Each red dot represents a cell expressing *Tmem63* mRNA. The single-cell RNA sequencing datasets are from previous studies[66,67] and are available in the Fly Cell Atlas repository.

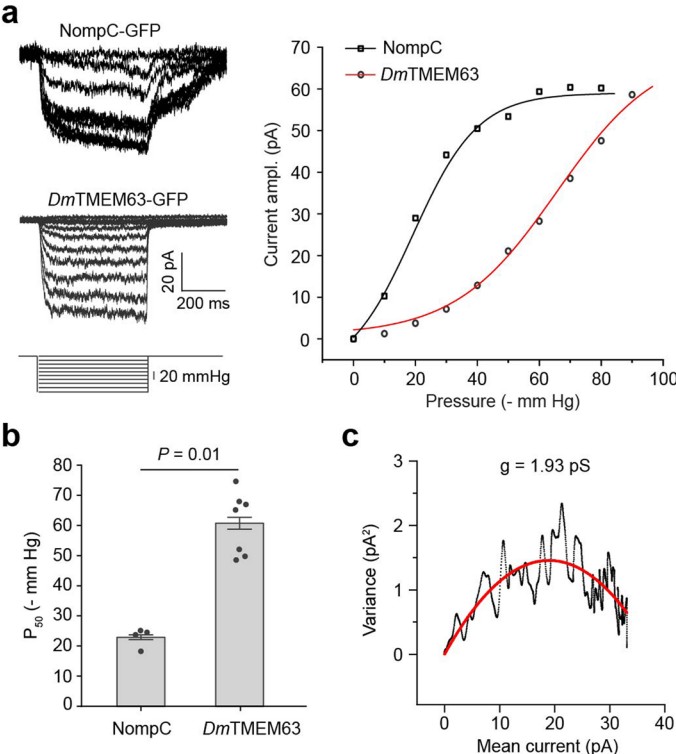

**Extended Data Fig. 3 | *Dm*TMEM63 is a high-threshold mechanosensitive channel with small conductance. a**, Representative current traces (left) and I-pressure curves fitted with the Boltzmann equation (right) of the NompC-GFP and *Dm*TMEM63-GFP channel proteins. **b**, Quantification of the pressure for half-maximal activation ($P_{50}$) of NompC (n = 4 cells) and *Dm*TMEM63 (n = 7 cells). Data shown are mean ± s.e.m. Statistical significance was determined by two-sided Mann-Whitney Test. **c**, Noise analysis of the fluctuations of the *Dm*TMEM63 currents. The variance of the current traces shown is plotted against the current amplitudes and fitted with $\sigma^2 = i \cdot I - I^2/N$. The obtained fitting parameters were i = -0.154 pA and N = 247 channels. So, the single-channel conductance is 1.93 pS at a holding potential of -80 mV. Numerical data is available as source data.

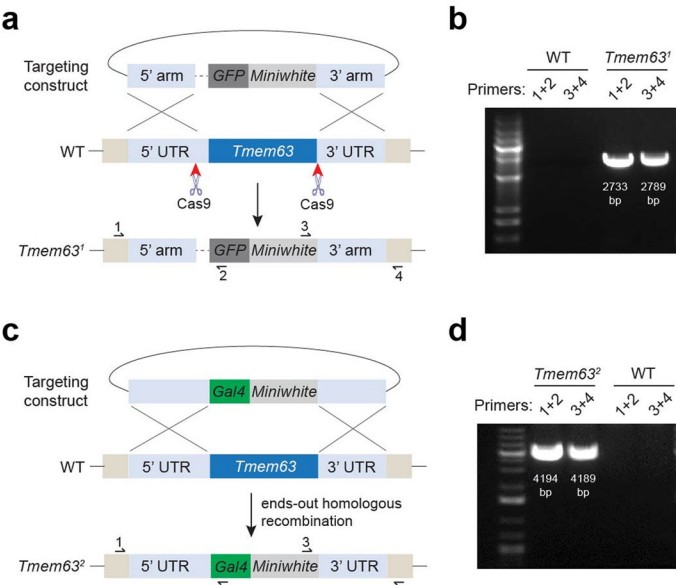

**Extended Data Fig. 4 | Generation of *Tmem63* knock-out flies. a**, *Tmem63¹* null mutant was generated through CRISPR-Cas9 mediated genome editing. The coding sequence of *Tmem63* was replaced with coding sequences of GFP and mini white. The GFP was not expressed in this line possibly due to a deletion in 5' UTR (shown as dashed lines). Arrowheads indicate cleavage sites of Cas9. **b**, PCR validation of *Tmem63¹* allele. **c**, *Tmem63²* null mutant was generated by ends-out homologous recombination. The coding sequence of *Tmem63* was replaced with Gal4 and mini white. The GAL4 was expressed in this line under the control of the native promoter of *Tmem63*. **d**, PCR validation of *Tmem63²* allele. The images in **b** and **d** are representatives of three independent experiments. The locations of the primers in **b** and **d** are shown in **a** and **c**, respectively. The sizes of the PCR fragments are indicated below the DNA bands. Unprocessed gels are available as source data.

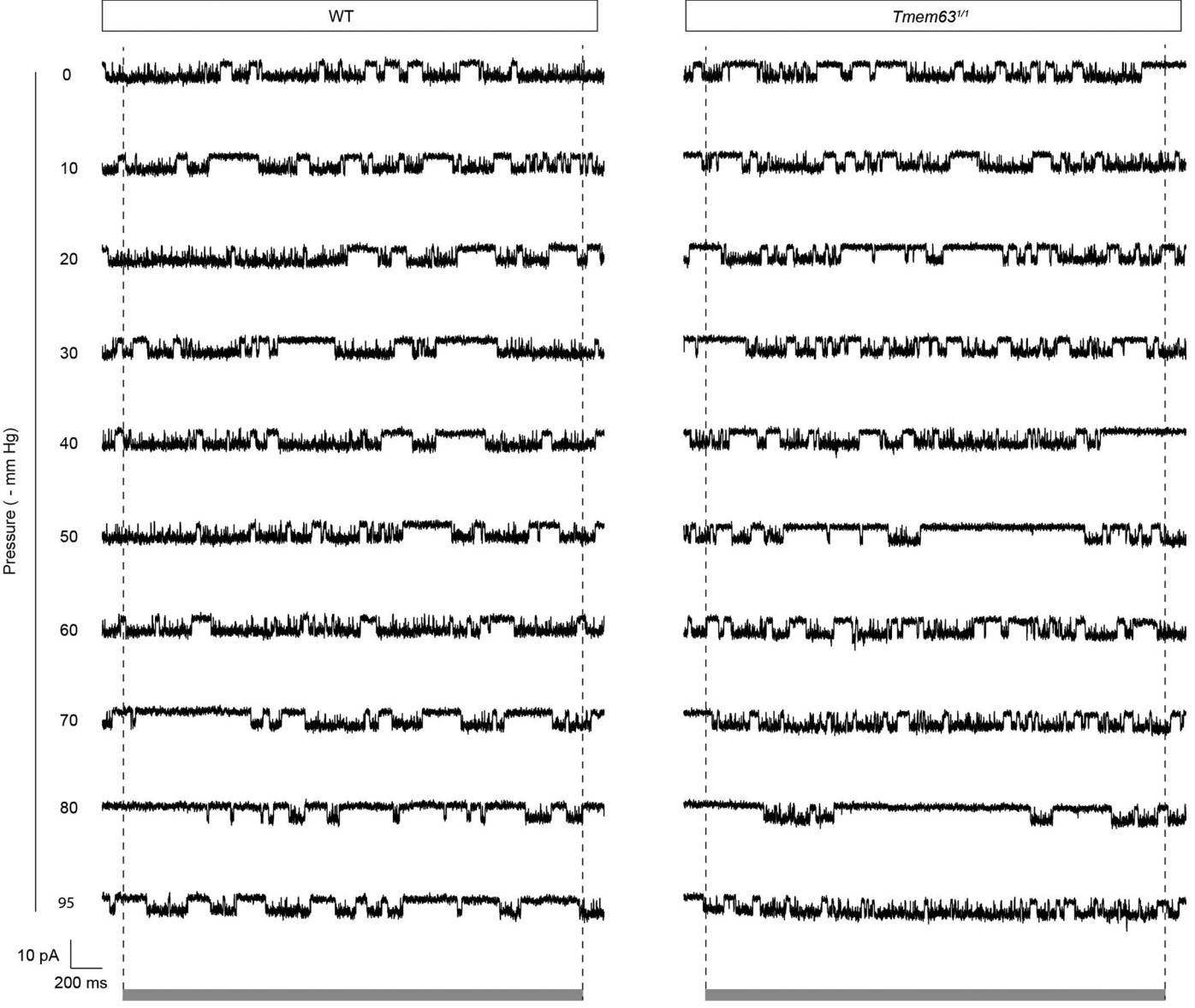

**Extended Data Fig. 5 | The native *Dm*TMEM63 currents were not detected in recordings from the plasma membranes of the fat-body cells.** Representative current traces of the fat-body plasma membranes in response to negative pressure in the cell-attached patch-clamp configuration. Fat bodies were dissected from wild-type (WT) and *Tmem63* mutant larvae. Grey bars indicate time duration of the pressure clamp.

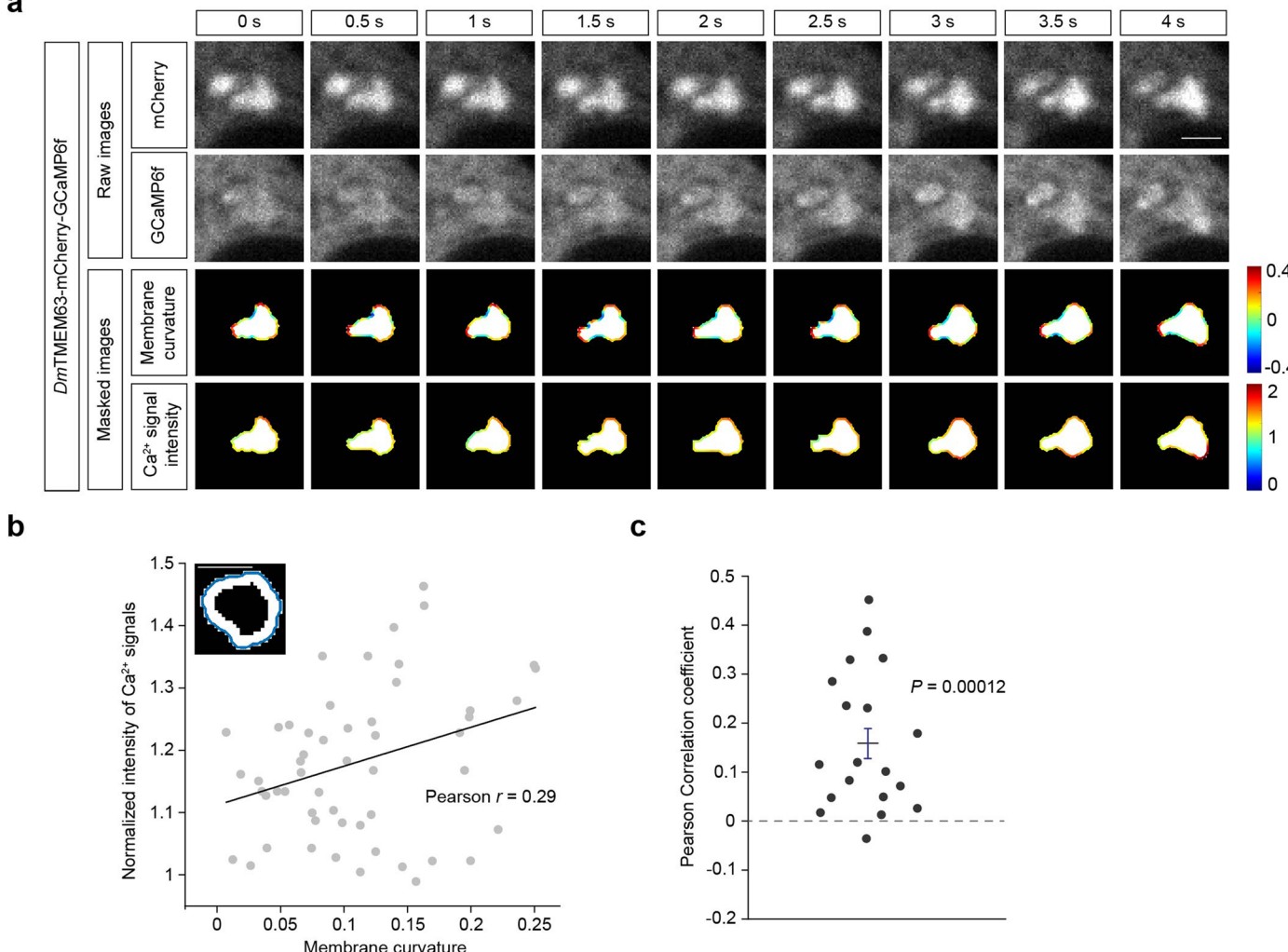

**Extended Data Fig. 6 | *Dm*TMEM63 channel activity correlates with lysosomal morphological remodeling. a**, Representative time-lapse images showing the lysosomal morphologies (mCherry) and lysosomal calcium signals (GCaMP6) in S2 cells expressing *Dm*TMEM63-mCherry-GCaMP6f. The heat maps of the masked images exhibit the membrane curvature and the intensity of Ca²⁺ signals of a lysosome. Scale bar, 1 μm. **b**, A positive correlation between membrane curvature and normalized Ca²⁺ signal (GCaMP6f fluorescent intensity divided by mCherry fluorescent intensity). The lysosome shown in inset was used for analysis. Blue line marks the boundary of the lysosome for calculating curvature

and the Ca²⁺ intensity measurement was carried out in the white region (for details, please see Methods). Pearson correlation coefficient (*r*) was used to determine the correlation between the curvature and Ca²⁺ signal. Each dot represents a correlation value of a segment of the lysosomal periphery at a specific timepoint. **c**, Group data of the Pearson correlation coefficient between the membrane curvature and intensity of Ca²⁺ signals. n = 19 lysosomes. Data shown are mean ± s.e.m. Statistical significance was determined by one-sample two-sided student's *t*-test. Scale bars, 1 μm (**a**, **b**). Numerical data is available as source data.

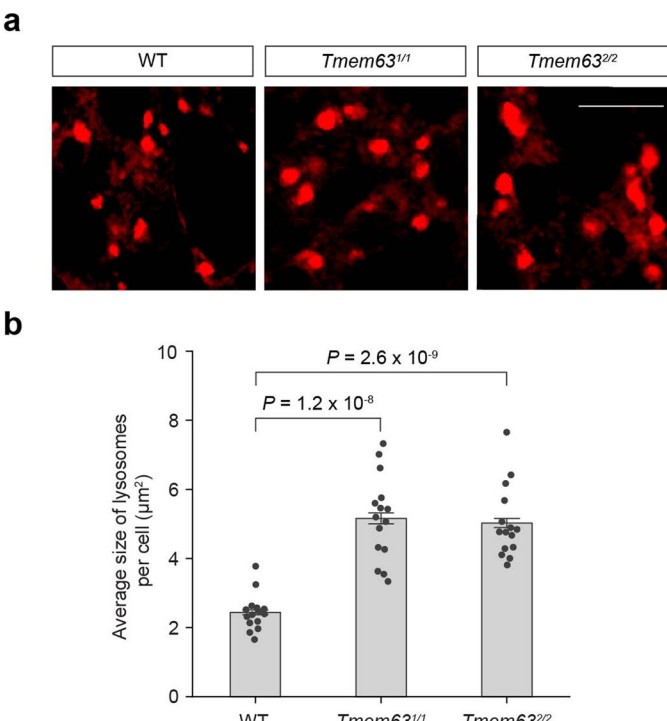

**Extended Data Fig. 7 | Increased lysosomal size in *Tmem63* mutants.**
Representative images (**a**) and quantifications (**b**) of lysosomes (labelled by
LysoTracker-red dye) in the fat bodies of the mid-L3 wild-type (WT) or *Tmem63*
mutant larvae under 4 h starved conditions. Scale bars, 10 μm. n = 15 cells per
group; data shown are mean ± s.e.m. Statistical significance was determined by
two-sided Student's *t*-test. Numerical data is available as source data.

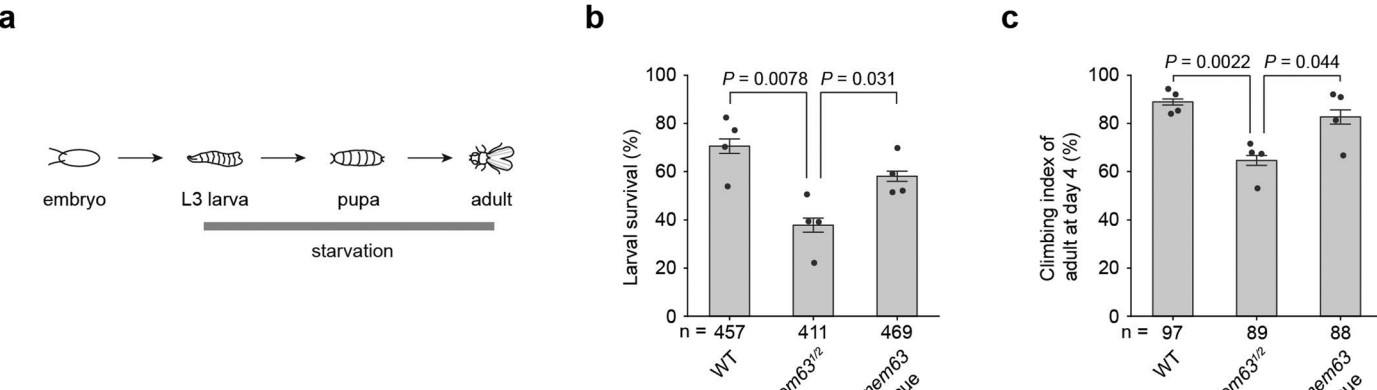

**Extended Data Fig. 8 | *Tmem63* mutant flies are vulnerable to starvation.**
**a**, Schematic showing the starvation stress treatment starting from the third-stage larvae (L3). **b**, Percentage of viable larvae for wild type (WT), *Tmem63* mutants, and *Tmem63* mutants re-expressing the *Tmem63* gene (*Tmem63* rescue). **c**, Climbing activities of young adult flies at day 4. All data shown are mean ± s.e.m. Statistical significance was determined by two-sided Student's *t*-test (**b**, **c**). The numbers of animals are shown beneath the bars. The fly genotypes in **b** and **c** are indicated in Supplementary Table 1. Numerical data are available as source data.

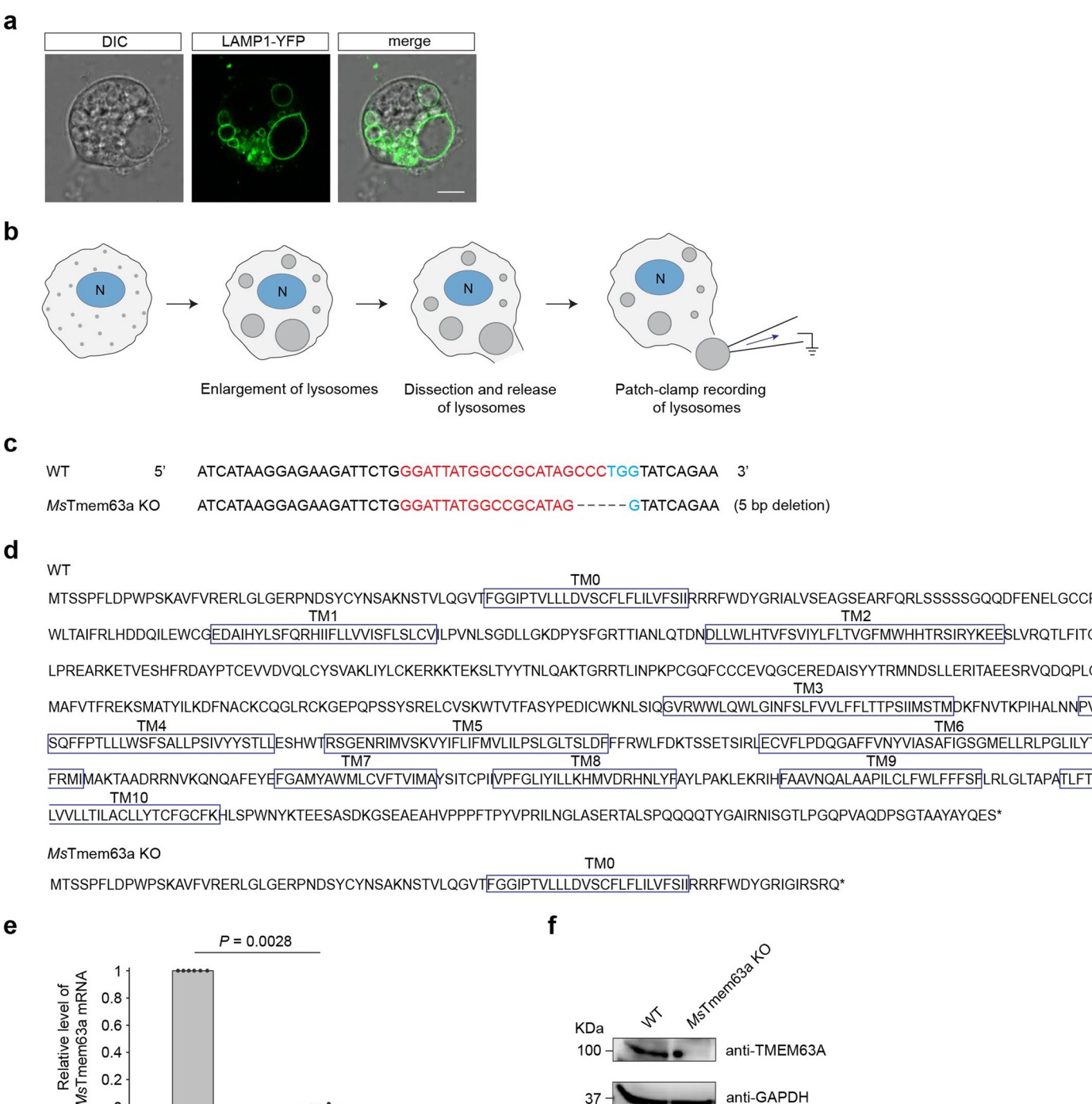

**Extended Data Fig. 9 | Lysosomal recordings and CRISPR-mediated gene disruption in N2a cells. a**, Enlarged lysosomes (LAMP1-YFP-positive intracellular vesicles) in N2a cells. Scale bar, 5 μm. **b**, Schematic of the experimental procedure for patch-clamp recordings on lysosomes of N2a cells. Lysosomes are shown as grey vesicles. N, cell nucleus. **c**, DNA sequences of the Cas9-targeted segments in wild-type (WT) and *Ms*Tmem63a knock-out (KO) N2a cells. Guide RNA and protospacer adjacent motif (PAM) sequences are highlighted in red and blue, respectively. **d**, The deletion in the *Ms*Tmem63a gene results in a pre-mature stop codon of the *Ms*TMEM63A proteins in KO cells. **e**, The mRNA level of *Ms*Tmem63a in WT and KO cells. Data shown are mean ± s.e.m. Statistical significance was determined by two-sided Mann-Whitney test. n = 6 independent experiments. **f**, The *Ms*TMEM63A protein level in WT and KO cells. The results in **a** and **f** are representatives of three independent experiments. Numerical data and unprocessed blots are available as source data.

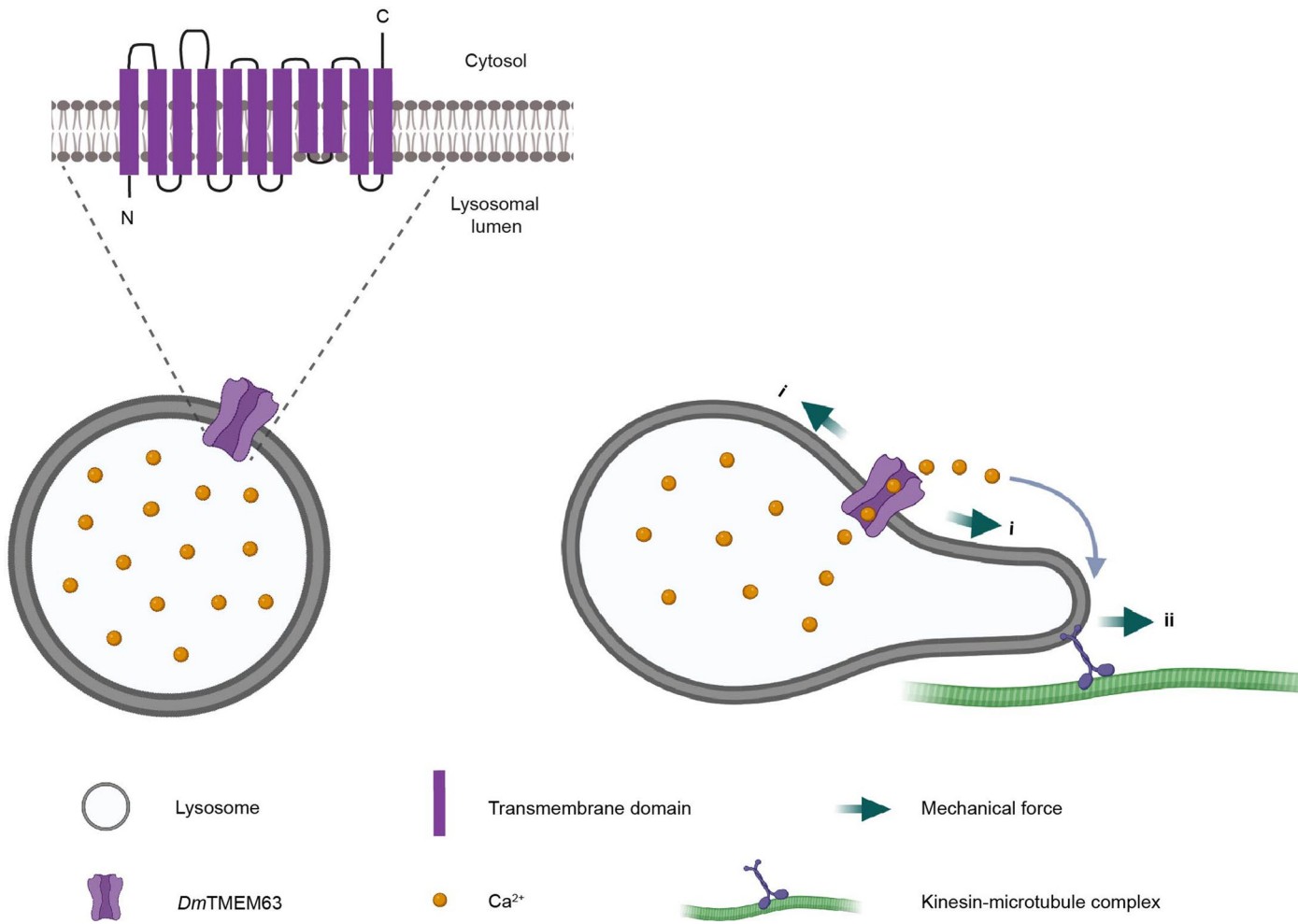

**Extended Data Fig. 10 | Model illustrating the proposed role of *Dm*TMEM63-mediated lysosomal mechanotransduction.** Left, the lysosomal membrane harbors the *Dm*TMEM63 protein, a mechanosensitive cationic channel. Right, the interplay between *Dm*TMEM63 channel activity and lysosomal dynamics. The lysosomal morphological remodeling alters the membrane curvature and surface tension (i) and cytoskeletal elements, such as kinesin-microtubule complex that drives lysosomal tubulation[23], exert mechanical forces on the lysosomal membrane (ii). These changes in mechanical state of the lysosomal membrane may activate the *Dm*TMEM63 channel and cause lysosomal $Ca^{2+}$ efflux which in turn regulates lysosomal remodeling and mobility. The model was created with BioRender.com.

# Reporting Summary

## Statistics

For all statistical analyses, confirm that the following items are present in the figure legend, table legend, main text, or Methods section.

| n/a | Confirmed | |
|---|---|---|
| ☐ | ☒ | The exact sample size (*n*) for each experimental group/condition, given as a discrete number and unit of measurement |
| ☐ | ☒ | A statement on whether measurements were taken from distinct samples or whether the same sample was measured repeatedly |
| ☐ | ☒ | The statistical test(s) used AND whether they are one- or two-sided *Only common tests should be described solely by name; describe more complex techniques in the Methods section.* |
| ☐ | ☒ | A description of all covariates tested |
| ☐ | ☒ | A description of any assumptions or corrections, such as tests of normality and adjustment for multiple comparisons |
| ☐ | ☒ | A full description of the statistical parameters including central tendency (e.g. means) or other basic estimates (e.g. regression coefficient) AND variation (e.g. standard deviation) or associated estimates of uncertainty (e.g. confidence intervals) |
| ☐ | ☒ | For null hypothesis testing, the test statistic (e.g. *F*, *t*, *r*) with confidence intervals, effect sizes, degrees of freedom and *P* value noted *Give P values as exact values whenever suitable.* |
| ☒ | ☐ | For Bayesian analysis, information on the choice of priors and Markov chain Monte Carlo settings |
| ☒ | ☐ | For hierarchical and complex designs, identification of the appropriate level for tests and full reporting of outcomes |
| ☐ | ☒ | Estimates of effect sizes (e.g. Cohen's *d*, Pearson's *r*), indicating how they were calculated |

*Our web collection on statistics for biologists contains articles on many of the points above.*

## Software and code

Policy information about availability of computer code

| Data collection | Confocal images were taken by the Olympus FV3000 with the FV31S-SW (version 2.6) software or Leica SP8 with the LAS X (version 4.1.1) software. Calcium imaging data were taken under ORCA-Flash4.0 Hamamatsu digital camera (version C13440) paired with Olympus IX73 microscope with Micro-manager (version 2.0.0) software or Olympus SpinSR spinning disk con-focal microscope with the cellSens (version 4.2) software. Electrophysiological recordings were performed with an Axopatch 700B amplifier and a Digidata 1440A or Digidata 1550B digitizer (Molecular Devices). Electrophysiology data were acquired by pClamp and Clampfit (Molecular Devices, version 10.4) softwares. |
|---|---|
| Data analysis | Origin 2021 was utilized for statistical analysis and generating graphs. Image J Fiji 2.1.0 was used for image quantification. For co-localization quantification, Pearson's correlation coefficient was quantified by Coloc2 plugin in ImageJ. Phylogenetic tree was constructed by the Phylogeny tool at EMBL-EBI. The analysis of correlation between lysosomal curvature and calcium intensity was performed in Matlab (version R2020a) with codes available at Github (https://github.com/weichensteven/lysosome_curvature_calcium_correlation_analysis). |

For manuscripts utilizing custom algorithms or software that are central to the research but not yet described in published literature, software must be made available to editors and reviewers. We strongly encourage code deposition in a community repository (e.g. GitHub). See the Nature Portfolio guidelines for submitting code & software for further information.

# Data

Policy information about availability of data

All manuscripts must include a data availability statement. This statement should provide the following information, where applicable:
- Accession codes, unique identifiers, or web links for publicly available datasets
- A description of any restrictions on data availability
- For clinical datasets or third party data, please ensure that the statement adheres to our policy

All data of this study are available in the main text or the extended materials with with numeric data of graphs and uncropped scans of all blots and gels shown in Source Data files. The genotypes of the animals used in figures and the sequences of oligonucleotides used in Methods are listed in Supplementary Table 1 and 2, respectively. For Extended Data Fig. 2b, datasets analyzed are available in the Fly Cell Atlas repository with the following weblink https://scope.aertslab.org/#/FlyCellAtlas/*/welcome. The two papers related to the dataset were cited in the Methods references of the manuscript.

## Human research participants

Policy information about studies involving human research participants and Sex and Gender in Research.

| | |
|---|---|
| Reporting on sex and gender | N/A |
| Population characteristics | N/A |
| Recruitment | N/A |
| Ethics oversight | N/A |

Note that full information on the approval of the study protocol must also be provided in the manuscript.

# Field-specific reporting

Please select the one below that is the best fit for your research. If you are not sure, read the appropriate sections before making your selection.

☒ Life sciences          ☐ Behavioural & social sciences          ☐ Ecological, evolutionary & environmental sciences

For a reference copy of the document with all sections, see nature.com/documents/nr-reporting-summary-flat.pdf

# Life sciences study design

All studies must disclose on these points even when the disclosure is negative.

| | |
|---|---|
| Sample size | No statistical method was used to predetermine sample size. Sample sizes were based on current standard in the field and previous studies (PMID: 23222543, PMID: 25959678, PMID: 30382938, PMID: 35780140). The sample sizes used in this paper are indicated in the corresponding figures or figure legends. |
| Data exclusions | The total number of the tested flies was counted after climbing assays, if there are dead flies after the experiments, the dead flies will not be counted. |
| Replication | All experiments were performed at least three times. All attempts at replication were successful. |
| Randomization | Animals or cells were allocated based on genotypes or treatments, which were described in the corresponding figure legends. Images were obtained randomly and were taken from random regions of prepared samples. |
| Blinding | The investigators who collected and analyzed the image data were blinded to group allocation. The investigators were blinded to the genotypes during the behavioural assays. Since only one single investigator typically performed all procedures of the electrophysiological experiments, the investigator was not blinded to the group allocation in electrophysiology. |

# Reporting for specific materials, systems and methods

We require information from authors about some types of materials, experimental systems and methods used in many studies. Here, indicate whether each material, system or method listed is relevant to your study. If you are not sure if a list item applies to your research, read the appropriate section before selecting a response.

## Materials & experimental systems

| n/a | Involved in the study |
|---|---|
| ☐ | ☒ Antibodies |
| ☐ | ☒ Eukaryotic cell lines |
| ☒ | ☐ Palaeontology and archaeology |
| ☐ | ☒ Animals and other organisms |
| ☒ | ☐ Clinical data |
| ☒ | ☐ Dual use research of concern |

## Methods

| n/a | Involved in the study |
|---|---|
| ☒ | ☐ ChIP-seq |
| ☒ | ☐ Flow cytometry |
| ☒ | ☐ MRI-based neuroimaging |

## Antibodies

**Antibodies used**

The primary antibodies used in the immunostaining experiments were: Mouse anti-GFP antibody (Roche, # 11814460001); Chicken anti-mCherry antibody (Novus Biologicals, # NBP2-25158); Mouse anti-BRP antibody (Developmental Studies Hybridoma Bank, # nc82). The secondary antibodies for immunostaining were anti-mouse labeled by Alexa 488 (ThermoFisher # A28175), or Cy3 (Jackson ImmunoResearch Labs # 115-165-146) and anti-chicken labeled by Alexa 647 (Jackson ImmunoResearch Labs, # 103-605-155). The primary antibody used in the Western blots were: Mouse anti-GFP antibody (Roche, # 11814460001); Rabbit anti-mCherry antibody (Abcam, # ab167453); Rabbit anti-Ref2P antibody (Abcam, # ab178440); Rabbit anti-TMEM63A antibody (Novus Biologicals, # NBP2-57359); Mouse anti-GAPDH antibody (Proteintech, # 60004-1-Ig); Mouse anti-tubulin antibody (Sigma, # 9026). The secondary antibodies for the Western blots were anti-mouse HRP antibody (Jackson ImmunoResearch Labs, # 115-035-146); anti-rabbit HRP antibody (Jackson ImmunoResearch Labs, # 111-035-144).

**Validation**

(1) Mouse anti-GFP antibody (Roche, # 11814460001; WB: 1:2000; IF: 1:500; https://www.sigmaaldrich.com/US/en/product/roche/11814460001#product-documentation)
(2) Chicken anti-mCherry antibody (Novus Biologicals, # NBP2-25158; IF: 1:300; https://www.novusbio.com/products/mcherry-antibody_nbp2-25158).
(3) Rabbit anti-mCherry antibody (Abcam, # ab167453; WB: 1:2000; https://www.abcam.com/products/primary-antibodies/mcherry-antibody-ab167453.html);
(4) Mouse anti-BRP antibody (1:100, Developmental Studies Hybridoma Bank, # nc82; IF: 1:100; https://dshb.biology.uiowa.edu/nc82).
(5) Rabbit anti-Ref2P antibody (Abcam, # ab178440; WB: 1:500; https://www.abcam.com/products/primary-antibodies/ref2p-antibody-ab178440.html).
(6) Mouse anti-GAPDH antibody (Proteintech, # 60004-1-Ig; WB: 1:2000; https://www.ptglab.com/products/GAPDH-Antibody-60004-1-Ig.htm);
(7) Mouse anti-tubulin antibody (Sigma, # 9026; WB: 1:2000; https://www.sigmaaldrich.com/US/en/product/sigma/t9026).
(8) Rabbit anti-TMEM63A antibody (Novus Biologicals, # NBP2-57359; WB: 1:200; https://www.novusbio.com/products/tmem63a-antibody_nbp2-57359). This antibody was also validated by experiments in this paper, because TMEM63A proteins could be detected through this antibody in N2a WT cells expressing endogenous TMEM63A (Fig. 6b) but not detected in the MsTmem63-KO cells (Extended Data Fig. 9f).

## Eukaryotic cell lines

Policy information about cell lines and Sex and Gender in Research

**Cell line source(s)**

The following cell lines were from ATCC: S2 cells (ATCC, # CRL-1963), HEK293 cells (ATCC, # CRL-1573), HEK293T cells (ATCC, # CRL-3216), and SH-SY5Y cells (ATCC, # CRL-2266). BV-2 cells were obtained from Dr. Li Gan lab and originally from InterLab Cell Line Collection, Banca Biologica e Cell Factory (ICLC, # ATL03001). N2a cells were obtained from Dr. Nevan Krogan lab and originally from ATCC (ATCC, # CCL-131).

**Authentication**

Cell line authentication was performed by the supplier through STR. In addition, S2 cells, HEK293 cells and N2a cells have distinct morphology and growing rate, which was used to distinguish the cell types during cell maintenance.

**Mycoplasma contamination**

Cell lines were negative for mycoplasma through PCR-based mycoplasma detection.

**Commonly misidentified lines**
(See ICLAC register)

No commonly misidentified cell lines were used.

## Animals and other research organisms

Policy information about studies involving animals; ARRIVE guidelines recommended for reporting animal research, and Sex and Gender in Research

**Laboratory animals**

(1) Fruit flies (Drosophila melanogaster) was used as the laboratory animal. Larvae or adults were used with the developmental stages or ages indicated in figures or figure legends.
(2) The following fly strains were obtained from the Bloomington Stock Center: stock # 39668 (UAS-spin.myc-GFP), stock # 7011 (Cg-GAL4), stock # 48183 (GMR51B08-GAL4), stock # 32194 (UAS-mCD8-GFP) and stock # 58772 (UAS-Piezo-GFP).
(3) The following stocks were obtained from other labs: UAS-GFP-LAMP1, Cg-GAL4 UAS-GFP-LAMP1.
(4) The following transgenic flies were generated in this study by P-element-mediated germline transformation or phiC31-mediated

integration: Tmem63-Gal4, UAS-Tmem63-GFP, UAS-Tmem63.
(5) The Tmem63 mutant flies were generated through ends-out homologous recombination or CRISPR/Cas9 technology.
(6) The DmTmem63-GFP and DmTmem63-mCherry knock-in flies were generated via CRISPR/Cas9 system.

| | |
|---|---|
| Wild animals | This study did not involve wild animals. |
| Reporting on sex | none |
| Field-collected samples | none |
| Ethics oversight | No ethical approval was required for the fly work. |

Note that full information on the approval of the study protocol must also be provided in the manuscript.

