## [Peer Review File · Nature Cell Biology]

Peer Review Information

Journal: Nature Cell Biology

Manuscript Title: Drosophila TMEM63 and mouse TMEM63A are lysosomal mechanosensory ion channels

Corresponding author name(s): Professor Yuh Nung Jan

Editorial Notes:

**Redactions –
unpublished data**

Reviewer Comments & Decisions:

Decision Letter, initial version:

Dear Dr Jan,

Thank you for submitting your manuscript, "The Drosophila TMEM63 channel is a lysosomal mechanosensor", to Nature Cell Biology. It has now been seen by 4 referees, who are experts in ion channels (Referee #1); ion regulation at lysosomes (Referee #2); ion channels and mechanobiology (Referee #3); and Drosophila, mechanobiology (Referee #4). As you will see from their comments (attached below), they found this work of potential interest but have raised substantial concerns, which in our view would need to be addressed with considerable revisions before we can consider publication in Nature Cell Biology.

Nature Cell Biology editors discuss the referee reports in detail within the editorial team, including the chief editor, to identify key referee points that should be addressed with priority, as opposed to requests that are beyond the scope of the current study. To guide the scope of the revisions, I have listed these points below. Our standard revision period is six months, and we are committed to providing a fair and constructive peer-review process, so please feel free to contact me if you would like to discuss any of the referee comments further or if you anticipate any issues or delays addressing the reviews.

In particular, in our view, it would be essential to address the following points:

-- all reviewers commented on the need to strengthen the evidence of functional conservation

between *Drosophila* and human TMEM63/TMEM63A and we agree that these points should be addressed experimentally:

Rev#1 major point #1

Rev#2 point #3

Rev#3 paragraph starting with "Rescue experiments with HsTMEM63a/b/c..."

Rev#4 point #5

-- additional studies teasing apart the function of the channel at lysosomes, as opposed to the plasma membrane, and further characterizing its behavior at lysosomes would bolster the claim of mechanosensitivity at lysosomes:

Rev#1 minor points #3; #6; #7

Rev#2 points #1-2

Rev#3 points #1-2-3-4

We also recommend that you strengthen the phenotypic studies in cells and in vivo as per Rev#4's points #3-4 and if possible explore Ca²⁺ dynamics from lysosomes as per Rev#4's suggestion in point #6.

-- Finally, please pay close attention to our guidelines on statistical and methodological reporting (listed below) as failure to do so may delay the reconsideration of the revised manuscript. In particular please provide:

We would be happy to consider a revised manuscript that would satisfactorily address these points, unless a similar paper is published elsewhere, or is accepted for publication in *Nature Cell Biology* in the meantime.

- ensure that it conforms to our format instructions and publication policies (see below and <https://www.nature.com/nature/for-authors>).

- provide a point-by-point rebuttal to the full referee reports verbatim, as provided at the end of this letter.

- provide the completed Reporting Summary (found here <https://www.nature.com/documents/nr-reporting-summary.pdf>). This is essential for reconsideration of the manuscript will be available to editors and referees in the event of peer review. For more information see <http://www.nature.com/authors/policies/availability.html> or contact me.

When submitting the revised version of your manuscript, please pay close attention to our [href="https://www.nature.com/nature-portfolio/editorial-policies/image-integrity">Digital Image Integrity Guidelines](https://www.nature.com/nature-portfolio/editorial-policies/image-integrity). and to the following points below:

Nature Cell Biology is committed to improving transparency in authorship. As part of our efforts in this direction, we are now requesting that all authors identified as 'corresponding author' on published papers create and link their Open Researcher and Contributor Identifier (ORCID) with their account on the Manuscript Tracking System (MTS), prior to acceptance. ORCID helps the scientific community achieve unambiguous attribution of all scholarly contributions. You can create and link your ORCID from the home page of the MTS by clicking on 'Modify my Springer Nature account'. For more information please visit www.springernature.com/orcid.

This journal strongly supports public availability of data. Please place the data used in your paper into a public data repository, or alternatively, present the data as Supplementary Information. If data can only be shared on request, please explain why in your Data Availability Statement, and also in the correspondence with your editor. Please note that for some data types, deposition in a public repository is mandatory - more information on our data deposition policies and available repositories appears below.

[Redacted]

We hope that you will find our referees' comments and editorial guidance helpful. Please do not hesitate to contact me if there is anything you would like to discuss. Thank you again for considering NCB for your work,

Best wishes,

Melina

Melina Casadio, PhD
Senior Editor, Nature Cell Biology
ORCID ID: <https://orcid.org/0000-0003-2389-2243>

Reviewers' Comments:

Reviewer #1:

Remarks to the Author:

In this interesting paper, the authors report the discovery of a novel lysosomal mechano-sensitive channel TMEM63 and its important functions in lysosomal morphology/physiology and the maintenance of synapses. Intracellular organelles including lysosomes and endosomes are highly mobile and are in frequent physical contact with other cellular compartments. As such, they almost certainly have to sense and respond to mechanical forces. However, how they do so has been very rarely studied; this is in contrast to the extensive studies, particular in the past decade, of how plasma membrane senses mechanical forces. This paper reports a set of quite convincing findings that the TMEM63 ion channel may fulfill the mechano-sensing function of lysosomes. By examining the localization of the "native" protein (with knock-in tags) in flies and directly patch clamp recording fly lysosomes (the first time this has been done), the authors demonstrate that the protein forms a mechanically-activated (MA) channel on the lysosome. It's interesting that TMEM63 appears to be the only major MA channel on the organelle since knocking it out diminishes the mechanical responses, which appears to be different from that of plasma membranes where several MA channels are present. Perhaps the even more important findings of the paper are the discovery of the functions of such an intracellular MA channel. Using fly mutants, the authors found that lysosomal morphology is compromised in the mutant, and there are defects in lysosomal degradation, loss of synapse, impairment in locomotive function and shortened life-span when TMEM63 is missing.

The findings are both novel and important. This is the first well-defined lysosomal MA channel (previous recordings have shown that the LRRC8A-implicated channel and TRPML2 are volume/osmolarity regulated and, for TRPML2, weakly MA though poorly defined). This is also the first study demonstrating the function of such a lysosomal MA channel at the whole-organismal level. In addition, the findings from *Drosophila* also appear to be applicable to humans, as a human TMEM63 homolog can partially rescue the fly mutant phenotype and a pathogenic mutant human TMEM63 is mislocalized.

I find that the studies are well designed and the results are convincing. The studies are comprehensive, from the identification of a novel lysosomal ion channel to the revealing of physiological roles at the organelle and whole animal levels. The findings will be of general interest to researchers in the fields of lysosomal biology, ion channel biophysics, neurobiology and neurodegenerative diseases. The paper is well written and easy to read, and the statistical studies are well performed. I only have one major comment and a few minor ones.

Major:

1) The authors appear to suggest that the functions of mammalian TMEM63s (hsTMEM63s) and the *Drosophila* dmTMEM63 are conserved. One piece of evidence is that human TMEM63A partially rescues

the fly phenotype. The degree of rescue (Fig. 6F), however, is moderate, and it's unclear whether the rescue is via the channel's lysosomal function or that at the plasma membrane. Given that hSTMEM163s have been shown to form plasma membrane channels, the authors should perhaps directly test with patch clamp recording whether hSTMEM163s also form lysosomal MA channels either in the S2 cells or, even better, in mammalian cells.

Minor:

2) Fig. 2A, the pH-sensitive GFP is fluorescent when fused to the C-terminus of the channel suggests that the C-terminus faces the cytosol, not the acidic lumen, consistent with the structure. The authors might want to point out this.

3) Fig. 3, how does the mechano sensitivity of dmTMEM63 compare to those of the plasma membrane channels such as NompC?

4) Fig. 3g, have the authors tried longer pressure pulses to test whether there is "inactivation"?

5) Fig. 4a,b: most of the other experiments used two independent alleles to minimize potential artifacts. It's unclear why only one allele was used in the studies of lysosomal sizes, a major conclusion.

6) Fig. 4f, h: is there increase in lysosomal MA channel currents in the over-expressor?

7) In Discussion, the authors state that "membrane, although we cannot exclude the possibility that its cell-surface expression is too sparse to be detected". Given the established plasma-membrane function of TMEM63 homologs from other animals, have the authors tested whether there is a native plasma-membrane dmTMEM63 current in cell-attached recording similar to that in Fig. 3a-c with over-expressed S2 cells?

8) reference: for references of lysosomal patch clamp recording, I'd also cite the Saito et al. (2007) JBC paper (PMCID #17609211). The Saito et al. paper was the first to describe the use of enlarged endolysosomes for patch clamp recording.

Reviewer #2:

Remarks to the Author:

In this manuscript by Li et al., the authors started to search for the organellar mechanosensory(s) by overexpressing known mechanosensitive (MS) channels in *Drosophila* S2 cells and found that DmTMEM63 proteins are specifically localized in the late endosomes and lysosomes. After confirming that the fluorescently-tagged endogenous DmTMEM63 proteins are also localized in the lysosomes, the authors performed electrophysiological recordings on released lysosomes from fat bodies to show that DmTMEM63 can indeed function as a lysosomal MS channel. Genetic inactivation of DmTMEM63 led to abnormal lysosomal morphology and subcellular localization. Tmem63 mutant flies exhibited lysosomal degradation defects, synaptic loss, progressive motor deficits, and shorter lifespan. Flies harboring a disease-causing mutation in DmTMEM63 (G540S) exhibited age-dependent climbing defects. The mutant phenotypes could be partially rescued by overexpressing HSTMEM63a, one of the three orthologs in human. Overall, this is an interesting study that may help advance our

understandings of the mechano-sensitivity of intracellular organelles. As TMEM63 appears to be expressed both at the plasma membrane and on the lysosomal membranes, the significance of the study can be improved if it can be shown that the lysosomal mechano-sensitivity is the primarily responsible for the observed cellular and animal level phenotypes (see below). In addition, additional control experiments are necessary to show that TMEM63 is both necessary and sufficient for lysosomal MS currents (see below). My specific comments are as follows:

1. While the overexpressed DmTMEM63 proteins are present on both plasma membrane and lysosomal membranes (see Fig. 1d and Fig. 3a), endogenous DmTMEM63 proteins appear to be expressed almost exclusively on the lysosomal membranes (see Fig. 2c and Fig. 3a). Can you perform lysosomal recordings in non-transfected S2 cells, and whole-cell recordings in fat cells, to show that endogenous DmTMEM63 is a lysosome-specific MS channel? In the heterologous overexpression systems, many lysosomal ion channels, for some reason, are also expressed at the plasma membrane, but their functions under physiological conditions are exclusively lysosomal. Alternatively, are there any lysosome-targeting motifs in TMEM63 that you can genetically manipulate to separate its roles in plasma membrane vs. lysosome? Note that some of the phenotypes, especially animal level ones, might be caused by TMEM63's non-lysosomal functions.
2. The kinetics (slow activation and no inactivation/adaptation) of the DmTMEM63 currents shown in Fig. 3a are quite different from those in the published work, e.g., by Murthy et al., 2018 *Elife*, which showed fast activation and inactivation. In Fig. 3g, h, the current amplitude is dependent on the pressure. I was assuming that Popen, but not single channel conductance, is increased by the increases in the mechanical pressure. Are there any reasons why no single channel currents can be seen in Fig. 3g? Was it because the single channel conductance was small, as predicted based on the lack of channel noise in the recordings. However, the channel noise in Fig. 3a was large, suggestive of large single channel conductance. To make Fig. 3g more convincing, it is necessary to provide additional controls. For instance, large lysosomal MS currents should be seen with DmTMEM63 overexpression. It might also be helpful to investigate whether HsTMEM63a, but not HsTMEM63b or HsTMEM63c can generate lysosomal MS currents in *Tmem63* KO cells, given the lack of rescue effects by HsTMEM63b or HsTMEM63c in Fig. 6f.
3. Along the line mentioned in #2, it might be helpful to study the effects of HsTMEM63 vs. HsTMEM63b/c expression on some of the lysosomal phenotypes, e.g., lysosomal size, observed in the *Tmem63* KO cells. This is important because the lysosomal phenotypes described in the current study, including the lysosomal size, are often seen in the cells carrying mutations in other lysosomal genes. I understand it may not be possible to prove that the loss of lysosomal mechanosensitivity is causal to the observed cellular phenotypes at this point, but any additional results from the correlation studies might be helpful. Note that DmTMEM63 might have non-lysosomal locations and possibly even non-MS functions.
4. In Fig. 2c, 2e, it is great to show that DmTMEM63 proteins are localized on the limited membranes of the lysosomes. A membrane protein can be localized in the lysosomes, either as a cargo, which would be detected in the lysosomal lumen, or as a lysosome-resident membrane protein, which would be detected on the limited membrane. DmTMEM63 is clearly in the latter case, and this important point should be emphasized in the revision.

Reviewer #3:

Remarks to the Author:

This interesting and elegant manuscript reports that the mechano-sensitive cationic channel DmTMEM63 is active at the lysosomal membrane of Drosophila cells. DmTMEM63 strongly influences lysosomal morphology, motor function and lifespan. Notably, DmTMEM63 mutant flies display important motor deficits and synaptic loss, recapitulating some of the human disease states associated with TMEM63 mutations. It is claimed that this function may be conserved in mammalian cells as HsTMEM63 partially rescues the behavioral defect of Tmem63 mutant flies.

As usual for this group the technical quality of this work is excellent, the manuscript is very well written and figures are nicely presented.

Although I find this study very interesting, I feel that there is major point that needs to be further addressed:

Rescue experiments with HsTMEM63a/b/c is very weak and not convincing (Fig. 6f), although those human channels are clearly present at the lysosomal membrane of HEK cells (Fig. 6a-c). Does it mean that TMEM63 plays a different role in mammalian cells and that the main function of HsTMEM63 isoforms is mainly at the plasma membrane? Did you try looking at HsTMEM63 localization in Drosophila cells (S2 cells for instance) and is it also found at the lysosomal membrane? I am really not convinced by the rescue data presented in this manuscript and perhaps it should be deleted. As an alternative, I suggest to explore a similar function for TMEM63 at the lysosomes from mammalian cells. At least show a similar mechanosensitive current in isolated lysosomes of mammalian cells (for instance HEK cells) and its possible link with TMEM63 (perhaps using a combination of siRNAs).

Additional points:

1) DmTMEM63-GFP induces mechano-sensitive cationic channels (reversing at 0 mV) recorded in the cell-attached patch clamp configurations. This is a bit confusing for the reader as this paper concerns the role of DmTMEM63 in lysosomes. Could you please add recordings of isolated lysosomes from S2 cells (the same way you did it for the fat body cells?).

2) It would be interesting to have an idea of the single channel conductance (I presume very small). Possibly a noise analysis could give you an estimate.

3) Recordings of DmTMEM63 at the lysosomal membrane in fat body cells is provided and this current is suppressed in the knock out lines. What happens to the mechano-sensitive current at the plasma membrane? Do you also observe a change (even though the plasma membrane expression of the GFP construct seems to be very weak at the cell surface as shown in Fig. 4a and 4c., but clearly overexpression shows an obvious expression at the plasma membrane as shown in Fig. 4e and 4g).

4) Can you please determine the calcium permeability of DmTMEM63? This is directly relevant to its physiological role in lysosomes.

5) It is mentioned in the discussion that DmTMEM63 may act as a dual osmo-mechano sensor. It would be interesting to demonstrate experimentally that DmTMEM63 at the lysosome membrane can also be activated by an hypo-osmotic shock.

General comments:

- A. Summary of the key results: Good
- B. Originality and significance: Novel and important for the field
- C. Data & methodology: validity of approach, quality of data, quality of presentation: Excellent but see the major point concerning the rescue experiments
- D. Appropriate use of statistics and treatment of uncertainties: Non parametric tests should be used for small n numbers (less than 15) as you cannot be sure about the normality of data.
- E. Conclusions: robustness, validity, reliability: Interesting discussion but again revise the rescue experiments
- F. Suggested improvements: experiments, data for possible revision: Explore the role of mammalian TMEM33A/B/C at the lysosomal membrane of mammalian cells
- G. References: appropriate credit to previous work? OK
- H. Clarity and context: lucidity of abstract/summary, appropriateness of abstract, introduction and conclusions. Good but see major point

Reviewer #4:

Remarks to the Author:

Report for The Drosophila TMEM63 channel is a lysosomal mechanosensor

A. Summary of the key results

In this manuscript Li et al., explore mechanosensing by intracellular organelles. While there has been recent excitement around the role of the nucleus as a mechanosensor for the cell, our knowledge of whether – and if so how – organelles sense and respond to force remains poorly understood. Here, they identify that the ion channel TMEM63 acts as a mechanosensor for the lysosome. Further, the authors demonstrate that the human form of TMEM63 can, at least partially, rescue function in Tmem63 fly mutant embryos, suggesting potential conservation.

B. Originality and significance: if not novel, please include reference

The results are important and have clear impact to cell biology. As far as I'm aware, the lysosome has not been shown to be mechanosensitive, making this an important insight.

C. Data & methodology: validity of approach, quality of data, quality of presentation

The methodology and results are generally of a high quality. I do have some suggestions for improvement below, but no major concerns.

D. Appropriate use of statistics and treatment of uncertainties

The statistical analysis is believable. Some experiments have limited replicas (e.g. Figure 5i) but these is reasonable given the long time course of this experiment.

E. Conclusions: robustness, validity, reliability

Overall, I found the results believable and well substantiated.

F. Suggested improvements: experiments, data for possible revision

I have some suggestions below.

1. Typically, modern papers have extremely dense figures with large amounts of information squeezed

in. Here, Figures 1 and 2 are somewhat of an exception. The localization shown in Figure 1 has quite a lot of redundant data that can be moved to SI. I suggest merging Figures 1 and 2 into a cleaner figure that focuses on the localization of TMEM63 to lysosomes. At the moment, the additional information hinders, rather than helps, the reader in identifying the key results.

2. In general, the presentation of the microscopy imaging is poor. The contrast and colour choices (green/red) used make it hard to see specific details such as localization. Fig 4 could be improved for clarity.

3. In Figure 4, it is shown that the phenotypic response depends on the fed/starvation state. Yet, this is not linked back to the mechanical forces facing the cells. How are forces altered under these different conditions? For example, lysosomes are shown to change area, but is this due to a change in the mechanical environment being sensed? In brief, what role do the authors posit for mechanical forces in the observations of Figure 4?

4. From a conceptual view (and on a similar theme to the above point), it is not clear that the phenotypes observed in Figure 5 are due to a loss of mechanosensitivity. TMEM63 may be acting through multiple pathways. Can larvae be raised under different environment stress conditions and their adult viability/climbing be assessed? One may imagine that the Tmem63 mutants would have even lower climbing index if derived from larvae that grew in a more stressed environment (as compared to changes in wt).

5. The authors should provide at least a hypothesis for why HsTMEM63A rescues (partially) whereas B and C variants do not. Is there something about the specific conserved domains that gives insight into the action of TMEM63 in *Drosophila*?

6. In the Discussion, the authors highlight that Ca dynamics should be recorded in conjunction with lysosome membrane remodelling. Though a full characterisation is not required, it does seem a clear omission (particularly given the prevalence of good Ca reporters) that such imaging is not performed here. It would clearly help link the purported changes in local mechanics (as the membrane changes) with action of TMEM63.

G. Clarity and context: lucidity of abstract/summary, appropriateness of abstract, introduction and conclusions

The paper is well written and the results are clear.

METHODS – Nature Cell Biology publishes methods online. The methods section should be provided as a separate Word document, which will be copyedited and appended to the manuscript PDF, and

incorporated within the HTML format of the paper.

Methods should be written concisely, but should contain all elements necessary to allow interpretation and replication of the results. As a guideline, Methods sections typically do not exceed 3,000 words. The Methods should be divided into subsections listing reagents and techniques. When citing previous methods, accurate references should be provided and any alterations should be noted. Information must be provided about: antibody dilutions, company names, catalogue numbers and clone numbers for monoclonal antibodies; sequences of RNAi and cDNA probes/primers or company names and catalogue numbers if reagents are commercial; cell line names, sources and information on cell line identity and authentication. Animal studies and experiments involving human subjects must be reported in detail, identifying the committees approving the protocols. For studies involving human subjects/samples, a statement must be included confirming that informed consent was obtained. Statistical analyses and information on the reproducibility of experimental results should be provided in a section titled "Statistics and Reproducibility".

All Nature Cell Biology manuscripts submitted on or after March 21 2016 must include a Data availability statement as a separate section after Methods but before references, under the heading "Data Availability". . For Springer Nature policies on data availability see <http://www.nature.com/authors/policies/availability.html>; for more information on this particular policy see <http://www.nature.com/authors/policies/data/data-availability-statements-data-citations.pdf>. The Data availability statement should include:

- Accession codes for primary datasets (generated during the study under consideration and designated as "primary accessions") and secondary datasets (published datasets reanalysed during the study under consideration, designated as "referenced accessions"). For primary accessions data should be made public to coincide with publication of the manuscript. A list of data types for which submission to community-endorsed public repositories is mandated (including sequence, structure, microarray, deep sequencing data) can be found here <http://www.nature.com/authors/policies/availability.html#data>.
- Unique identifiers (accession codes, DOIs or other unique persistent identifier) and hyperlinks for datasets deposited in an approved repository, but for which data deposition is not mandated (see here for details <http://www.nature.com/sdata/data-policies/repositories>).
- At a minimum, please include a statement confirming that all relevant data are available from the authors, and/or are included with the manuscript (e.g. as source data or supplementary information), listing which data are included (e.g. by figure panels and data types) and mentioning any restrictions on availability.
- If a dataset has a Digital Object Identifier (DOI) as its unique identifier, we strongly encourage including this in the Reference list and citing the dataset in the Methods.

We recommend that you upload the step-by-step protocols used in this manuscript to the Protocol Exchange. More details can found at www.nature.com/protocolexchange/about.

DISPLAY ITEMS – main display items are limited to 6-8 main figures and/or main tables for Articles, Resources, Technical Reports; and 5 main figures and/or main tables for Letters. For Supplementary

Information see below.

All imaging data should be accompanied by scale bars, which should be defined in the legend. Cropped images of gels/blots are acceptable, but need to be accompanied by size markers, and to retain visible background signal within the linear range (i.e. should not be saturated). The boundaries of panels with low background have to be demarked with black lines. Splicing of panels should only be considered if unavoidable, and must be clearly marked on the figure, and noted in the legend with a statement on whether the samples were obtained and processed simultaneously. Quantitative comparisons between samples on different gels/blots are discouraged; if this is unavoidable, it should only be performed for samples derived from the same experiment with gels/blots were processed in parallel, which needs to be stated in the legend.

The total number of Supplementary Figures (not including the “unprocessed scans” Supplementary Figure) should not exceed the number of main display items (figures and/or tables (see our Guide to Authors and March 2012 editorial <http://www.nature.com/ncb/authors/submit/index.html#suppinfo>; <http://www.nature.com/ncb/journal/v14/n3/index.html#ed>). No restrictions apply to Supplementary Tables or Videos, but we advise authors to be selective in including supplemental data.

GUIDELINES FOR EXPERIMENTAL AND STATISTICAL REPORTING

REPORTING REQUIREMENTS – We are trying to improve the quality of methods and statistics reporting in our papers. To that end, we are now asking authors to complete a reporting summary that collects information on experimental design and reagents. The Reporting Summary can be found here <https://www.nature.com/documents/nr-reporting-summary.pdf> If you would like to reference the guidance text as you complete the template, please access these flattened versions at <http://www.nature.com/authors/policies/availability.html>.

Author Rebuttal to Initial comments

Point-by-point response to the reviewers

We greatly appreciate all four reviewers' insightful comments on our work as well as their very helpful suggestions to improve our manuscript. Following their suggestions, we have performed many additional experiments and analysis and modified the figures and the manuscript accordingly. With the new data, we have confirmed and extended our conclusions from the original submission. Our point-by-point responses are detailed below.

Responses to Reviewer #1 (Reviewer's comments in italics):

Remarks to the Author:

In this interesting paper, the authors report the discovery of a novel lysosomal mechano-sensitive channel TMEM63 and its important functions in lysosomal morphology/physiology and the maintenance of synapses. Intracellular organelles including lysosomes and endosomes are highly mobile and are in frequent physical contact with other cellular compartments. As such, they almost certainly have to sense and respond to mechanical forces. However, how they do so has been very rarely studied; this is in contrast to the extensive studies, particular in the past decade, of how plasma membrane senses mechanical forces. This paper reports a set of quite convincing findings that the TMEM63 ion channel may fulfill the mechano-sensing function of lysosomes. By examining the localization of the "native" protein (with knock-in tags) in flies and directly patch clamp recording fly lysosomes (the first time this has been done), the authors demonstrate that the protein forms a mechanically-activated (MA) channel on the lysosome. It's interesting that TMEM63 appears to be the only major MA channel on the organelle since knocking it out diminishes the mechanical responses, which appears to be different from that of plasma membranes where several MA channels are present. Perhaps the even more important findings of the paper are the discovery of the functions of such an intracellular MA channel. Using fly mutants, the authors found that lysosomal morphology is compromised in the mutant, and there are defects in lysosomal degradation, loss of synapse, impairment in locomotive function and shortened life span when TMEM63 is missing.

The findings are both novel and important. This is the first well-defined lysosomal MA channel (previous recordings have shown that the LRRC8A-implicated channel and TRPML2 are volume/osmolarity regulated and, for TRPML2, weakly MA though poorly defined). This is also the first study demonstrating the function of such a lysosomal MA channel at the whole-organismal

level. In addition, the findings from *Drosophila* also appear to be applicable to humans, as a human TMEM63 homolog can partially rescue the fly mutant phenotype and a pathogenic mutant human TMEM63 is mislocalized.

I find that the studies are well designed and the results are convincing. The studies are comprehensive, from the identification of a novel lysosomal ion channel to the revealing of physiological roles at the organelle and whole animal levels. The findings will be of general interest to researchers in the fields of lysosomal biology, ion channel biophysics, neurobiology and neurodegenerative diseases. The paper is well written and easy to read, and the statistical studies are well performed. I only have one major comment and a few minor ones.

Major:

*1) The authors appear to suggest that the functions of mammalian TMEM63s (hsTMEM63s) and the *Drosophila* dmTMEM63 are conserved. One piece of evidence is that human TMEM63A partially rescues the fly phenotype. The degree of rescue (Fig. 6F), however, is moderate, and it's unclear whether the rescue is via the channel's lysosomal function or that at the plasma membrane. Given that hsTMEM63s have been shown to form plasma membrane channels, the authors should perhaps directly test with patch clamp recording whether hsTMEM63s also form lysosomal MA channels either in the S2 cells or, even better, in mammalian cells.*

Authors' response: In accordance with the reviewer's suggestion, we have performed new experiments in mammalian cells. Firstly, we found that N2a cells, a mouse neuroblastoma cell line, showed high-level expression of endogenous *MsTMEM63A* protein (Fig. 5b). We then performed patch-clamp recordings on the enlarged lysosomes of N2a cells and detected mechanosensitive (MS) currents from the lysosomal membrane (Fig. 5e). We further knocked out the *MsTmem63a* gene in N2a cells (Extended Data Fig. 10c-f) and found that the MS currents were largely abolished in *MsTmem63a*-KO cells (Fig. 5e). These results suggest that the mammalian lysosomes, just like *Drosophila* lysosomes, are intrinsically mechanosensitive, and TMEM63A forms a lysosomal MS channel in mammalian cells as well. Thus, we provide direct evidence to show the functional conservation between *Drosophila* and mammalian TMEM63 proteins.

These new data have been included in the revised manuscript (Fig. 5a-f, Extended Data Fig. 10 and Pages 12-13, lines 259-276). We also have added a sentence “Importantly, mouse TMEM63A mediates lysosomal mechanosensitivity in Neuro-2a cells, indicative of functional conservation in mammals” in the abstract of the revised manuscript.

Minor:

2) *Fig. 2A, the pH-sensitive GFP is fluorescent when fused to the C-terminus of the channel suggests that the C-terminus faces the cytosol, not the acidic lumen, consistent with the structure. The authors might want to point out this.*

Authors’ response: We have added the following sentence in the revised manuscript “Notably, the pH-sensitive GFP showed fluorescence in *Dm*TMEM63^{GFP} flies, which suggests the C-terminus of *Dm*TMEM63 faces to the cytosol, not the acidic lumen”. Please refer to Page 5, lines 107-109.

3) *Fig. 3, how does the mechano sensitivity of dmTMEM63 compare to those of the plasma membrane channels such as NompC?*

Authors’ response: We analyzed the pressure for half-maximal activation (P_{50}) of *Dm*TMEM63 and NompC. We found that the P_{50} value of *Dm*TMEM63 is larger than that of NompC (60.71 ± 3.95 versus 22.88 ± 1.58), suggesting *Dm*TMEM63 is less sensitive to mechanical stimuli than NompC. We have added the new data in the revised manuscript (Extended Data Fig. 3a, b and Page 6, lines 122-123).

4) *Fig. 3g, have the authors tried longer pressure pulses to test whether there is “inactivation”?*

Authors’ response: We tested a longer stimulus (1 second pulse at -50 mm Hg) in the lysosomal patch-clamp recording. We did not observe an apparent inactivation of the MS current (please see the trace below).

5) Fig. 4a,b: most of the other experiments used two independent alleles to minimize potential artifacts. It's unclear why only one allele was used in the studies of lysosomal sizes, a major conclusion.

Authors' response: To visualize lysosomes *in vivo*, we used Cg-Gal4, a fat-body-specific Gal4, to drive UAS-GFP-LAMP1 to label the fat-body lysosomes in wild type (WT) and *Tmem63*¹ flies. But this tissue-specific labeling cannot be used in the *Tmem63*² allele, because *Tmem63*² mutant has the Gal4 inserted into the *Tmem63* locus (please see Extended data Fig. 5c). So, we only used one allele in this experiment.

To further strengthen the conclusion, we have performed an additional experiment to visualize the lysosomes through LysoTracker staining which allows the usage of these two independent alleles. We found the lysosomal size is increased in both mutants. We have added the new data in the revised manuscript (Extended Data Fig. 8 and Page 9, lines 200-201). Please note that, in agreement with previous reports^{1,2}, we found the LysoTracker dye can only label acidic lysosomes in the starved larvae of L3 stage, but not the lysosomes in the fed larvae. We have therefore made this point in the Method Section (Page 28-29, lines 697-699), and only showed the LysoTracker-staining results of the fat body under starvation condition.

6) Fig. 4f, h: is there increase in lysosomal MA channel currents in the over-expressor?

Authors' response: We attempted to record the lysosomal MA currents of the fat bodies with *Tmem63* over-expression (*Tmem63*-OE). But we found it very difficult to form gigaseals between recording pipettes and the isolated lysosomes. As shown in the images below, the lysosomes isolated from *Tmem63*-OE cells are in irregular shapes (this is consistent with the observation in the fat body cells *in vivo*, please see Fig. 3g); their sectional areas are almost identical to, or even smaller than, the tips of the recording pipettes (the tips are typically 1 μm in diameter). In addition,

the lysosomes isolated from fat body appear to be insensitive to vacuolin-1, a compound used to enlarge mammalian lysosomes³. Thus, we were unable to record the lysosomal MA currents from the *Tmem63*-OE cells for these technical reasons. We have described the technical problem in the Method section of the revised manuscript (Page 25, lines 614-617).

7) In Discussion, the authors state that “membrane, although we cannot exclude the possibility that its cell-surface expression is too sparse to be detected”. Given the established plasma-membrane function of TMEM63 homologs from other animals, have the authors tested whether there is a native plasma-membrane *dm*TMEM63 current in cell-attached recording similar to that in Fig. 3a-c with over-expressed S2 cells?

Authors’ response: Following the reviewer’s suggestion, we have performed patch-clamp recordings to test whether there is a native *Dm*TMEM63 currents in the plasma membrane of the fat body cells. We did not detect the MA currents from the plasma membranes of either the WT or *Tmem63*-KO cells, suggesting that the endogenous *Dm*TMEM63 proteins are not localized or not functional in the plasma membranes of the fat body cells. These new data have been included in the revised manuscript (Extended Data Fig. 6 and Page 8, lines 167-174).

8) reference: for references of lysosomal patch clamp recording, I'd also cite the Saito et al. (2007) JBC paper (PMCID #17609211). The Saito et al. paper was the first to describe the use of enlarged endolysosomes for patch clamp recording.

Authors' response: We thank the reviewer for pointing out this original paper of the technique for patch clamp recording from enlarged endolysosomes. We have cited the paper in the Method section of the revised manuscript (Page 24, line 608 and Page 25, line 623).

Responses to Reviewer #2 (Reviewer's comments in italics):

In this manuscript by Li et al., the authors started to search for the organellar mechanosensory(s) by overexpressing known mechanosensitive (MS) channels in Drosophila S2 cells and found that DmTMEM63 proteins are specifically localized in the late endosomes and lysosomes. After confirming that the fluorescently-tagged endogenous DmTMEM63 proteins are also localized in the lysosomes, the authors performed electrophysiological recordings on released lysosomes from fat bodies to show that DmTMEM63 can indeed function as a lysosomal MS channel. Genetic inactivation of DmTMEM63 led to abnormal lysosomal morphology and subcellular localization. Tmem63 mutant flies exhibited lysosomal degradation defects, synaptic loss, progressive motor deficits, and shorter lifespan. Flies harboring a disease-causing mutation in DmTMEM63 (G540S) exhibited age-dependent climbing defects. The mutant phenotypes could be partially rescued by overexpressing HsTMEM63a, one of the three orthologs in human. Overall, this is an interesting study that may help advance our understandings of the mechano-sensitivity of intracellular organelles. As TMEM63 appears to be expressed both at the plasma membrane and on the lysosomal membranes, the significance of the study can be improved if it can be shown that the lysosomal mechano-sensitivity is the primarily responsible for the observed cellular and animal level phenotypes (see below). In addition, additional control experiments are necessary to show that TMEM63 is both necessary and sufficient for lysosomal MS currents (see below). My specific comments are as follows:

1. While the overexpressed DmTMEM63 proteins are present on both plasma membrane and lysosomal membranes (see Fig. 1d and Fig. 3a), endogenous DmTMEM63 proteins appear to be

expressed almost exclusively on the lysosomal membranes (see Fig. 2c and Fig. Fig. 3a). Can you perform lysosomal recordings in non-transfected S2 cells, and whole-cell recordings in fat cells, to show that endogenous DmTMEM63 is a lysosome-specific MS channel? In the heterologous overexpression systems, many lysosomal ion channels, for some reason, are also expressed at the plasma membrane, but their functions under physiological conditions are exclusively lysosomal. Alternatively, are there any lysosome-targeting motifs in TMEM63 that you can genetically manipulate to separate its roles in plasma membrane vs. lysosome? Note that some of the phenotypes, especially animal level ones, might be caused by TMEM63's non-lysosomal functions.

Authors' response: Here, the reviewer suggested two options to tease apart the actions of TMEM63 at lysosomes, as opposed to the plasma membrane. We appreciate the reviewer's insightful advice and comments. We have performed additional experiments to address these questions:

(1) We tried identifying a lysosome-targeting motif in *DmTMEM63*. The canonical lysosome-sorting signals include DXX[ML][VL] or YXXØ motifs located in the cytosolic domain of a lysosomal protein⁴. Based on these criteria, we identified several candidate motifs in *DmTMEM63*, and then mutated the key residues in the motifs to test how the localization patterns of these mutant proteins may be altered (please see the locations of the motifs/residues in panel **a** of the Figure below). Unfortunately, as shown in panel **b**, none of these mutant proteins are exclusively localized to the plasma membrane instead of lysosomes. Whereas the *DmTMEM63*^{Y715A}-GFP proteins showed increased fluorescence in the plasma-membrane of the fat body cells, the mutant proteins still had strong intracellular signals that co-localized with lysosomes. Thus, in our current efforts, we have not found a motif that could be manipulated to separate the role of *DmTMEM63* in the plasma membrane versus lysosomes.

Figure legend: **a**, The primary sequence of the *Dm*TMEM63 protein. The transmembrane domains are indicated in boxes. The candidate lysosome-targeting motifs are underlined with blue bars and the key residues are marked with arrows. **b**, The localization patterns of the WT and mutant *Dm*TMEM63-GFP proteins in the larval fat body. The proteins are expressed in the *Tmem63*^{-/-} background. The lysosomes are labelled with mCherry-LAMP1. Lines indicate a single cell within the fat-body tissues. Scale bar, 10 μ m.

(2) Following the reviewer's suggestion, we have performed patch-clamp recordings to evaluate the functions of native *Dm*TMEM63 channel in the plasma membrane versus lysosomes. We first tried lysosomal recordings in *Drosophila* S2 cells, but found the S2 cell lysosomes could not be enlarged after treatment of vacuolin-1, a compound commonly used to enlarge lysosomes for electrophysiological recordings³. The low sensitivity to vacuolin-1 of S2 cells was also reported in a previous study in which the authors stated they were unable to enlarge S2 cell lysosomes for patch-clamp recordings⁵. Thus, we did not have a chance to evaluate the currents of *Dm*TMEM63 in lysosomes of S2 cells. However, we examined whether there is a *Dm*TMEM63-mediated mechanosensitive (MS) current in the plasma membrane of the fat body. In contrast to the MS currents recorded from fat-body lysosomes, we did not observe any MS currents from the plasma membranes of either the WT or the *Tmem63*-KO fat body, suggesting the endogenous *Dm*TMEM63 proteins are not localized or not functional in the plasma membrane. This new result, together with the exclusive lysosomal localization of endogenous *Dm*TMEM63 (as noted by the reviewer) and its essential role for lysosomal mechanosensitivity, indicates that *Dm*TMEM63 is a lysosome-specific MS channel in the fat body cells. This new data have been added in the revised manuscript (Extended Data Fig. 6 and Page 8, lines 167-174).

2. The kinetics (slow activation and no inactivation/adaptation) of the DmTMEM63 currents shown in Fig.3a are quite different from those in the published work, e.g. by Murthy et al., 2018 Elife, which showed fast activation and inactivation. In Fig. 3g, h, the current amplitude is dependent on the pressure. I was assuming that Popen, but not single channel conductance, is increased by the increases in the mechanical pressure. Are there any reasons why no single channel currents can be seen in Fig. 3g? Was it because the single channel conductance was small, as predicted based on the lack of channel noise in the recordings. However, the channel noise in Fig. 3a was large, suggestive of large single channel conductance. To make Fig.3g more convincing, it is necessary to provide additional controls. For instance, large lysosomal MS currents should be seen with DmTMEM63 overexpression. It might also be helpful to investigate whether HsTMEM63a, but not HsTMEM63b or HsTMEM63c can generate lysosomal MS currents in Tmem63 KO cells, given the lack of rescue effects by HsTMEM63b or HsTMEM63c in Fig. 6f.

Authors' response: In response to the reviewer's comments and suggestions, we have performed new analyses and experiments:

(1) In our recordings, we did not observe an apparent inactivation of the *Dm*TMEM63 currents. A difference between our experiment and the previous work (Murthy et al. 2018, *elife*) is the expression system, that is, we expressed *Dm*TMEM63 in the *Drosophila* S2 cells and they expressed it in the mammalian HEK293 cells. It is conceivable that the temperature (S2 is maintained in 25°C while HEK293 is in 37°C) and/or species-specific cellular factors differentially modulate the folding, trafficking and/or kinetics of the channel protein. For example, Piezo1, a mechanosensitive channel, showed variable kinetics in different expression systems^{6,7}.

(2) We did not detect the single-channel currents in our recordings, which is consistent with a recent study reporting that the single-channel currents of TMEM63 proteins are too small to be resolved⁸. Indeed, through noise analysis, we have estimated the single-channel conductance of *Dm*TMEM63 (~1.93 pS), which is very small. We have added the new data in the revised manuscript (Extended Data Fig. 3c and Page 6, line 124).

(3) Lysosomes isolated from the cells expressing *Dm*Tmem63 or *Hs*Tmem63a have irregular shapes (as shown in the images below), which makes it very difficult to form gigaohm seal between the lysosomal membrane and the recording pipette (please also see our response to the minor point 6 of Reviewer #1). We attempted to perform recordings on these lysosomes but did not obtain reliable data. This technical problem is indicated in the Method section of the revised manuscript (Page 25, lines 614-617).

(4) As an alternative way, we characterized the MS currents from the enlarged lysosomes of the N2a cells, a mouse cell line. The MS currents were detected in the lysosomes of the wild-type N2a cells (Fig. 5e, f), but abolished in the *MsTmem63a*-KO cells (Fig. 5e, f), suggesting that the mouse TMEM63A mediates the lysosomal mechanosensitivity in N2a cells. Notably, we observed large lysosomal currents in cells over-expressing the *MsTMEM63A* (Fig. 5e, f). These results demonstrate that *MsTMEM63A* channel is both necessary and sufficient for the lysosomal MS currents. We have added the new results in the revised manuscript (Fig. 5e, f, Extended Data Fig. 10 and Pages 12-13, lines 268-276).

3. Along the line mentioned in #2, it might be helpful to study the effects of *HsTMEM63a* vs. *HsTMEM63b/c* expression on some of the lysosomal phenotypes, e.g. lysosomal size, observed in the *Tmem63* KO cells. This is important because the lysosomal phenotypes described in the current study, including the lysosomal size, are often seen in the cells carrying mutations in other lysosomal genes. I understand it may not be possible to prove that the loss of lysosomal mechanosensitivity is causal to the observed cellular phenotypes at this point, but any additional results from the correlation studies might be helpful. Note that *DmTMEM63* might have non-lysosomal locations and possibly even non-MS functions.

Authors' response:

[Redacted]

In this new version of the manuscript, we have mainly focused on the *MsTMEM63* instead of *HsTMEM63* (we used the mouse cell line N2a as the expression system to study mammalian TMEM63, as shown in Fig. 5). We also have removed the data regarding the ectopic rescue experiment with *HsTMEM63s* (as suggested by Reviewer #3 in the major point) [Redacted].

4. In Fig. 2c, 2e, it is great to show that DmTMEM63 proteins are localized on the limited membranes of the lysosomes. A membrane protein can be localized in the lysosomes, either as a cargo, which would be detected in the lysosomal lumen, or as a lysosome-resident membrane protein, which would be detected on the limited membrane. DmTMEM63 is clearly in the latter case, and this important point should be emphasized in the revision.

Authors' response: We agree with the reviewer about the importance of this point and we have added the following sentence in the revised manuscript “the membrane-located patterns of *DmTMEM63* demonstrated that *DmTMEM63* is a lysosome-resident protein rather than a cargo that is to be degraded in the lysosomal lumen”. Please refer to the Page 5, lines 109-111.

Responses to Reviewer #3 (Reviewer's comments in italics):

This interesting and elegant manuscript reports that the mechano-sensitive cationic channel DmTMEM63 is active at the lysosomal membrane of Drosophila cells. DmTMEM63 strongly influences lysosomal morphology, motor function and lifespan. Notably, DmTMEM63 mutant flies display important motor deficits and synaptic loss, recapitulating some of the human disease states associated with TMEM63 mutations. It is claimed that this function may be conserved in mammalian cells as HsTMEM63 partially rescues the behavioral defect of Tmem63 mutant flies.

As usual for this group the technical quality of this work is excellent, the manuscript is very well written and figures are nicely presented. Although I find this study very interesting, I feel there is a major point that needs to be further addressed:

Rescue experiments with HsTMEM63a/b/c is very weak and not convincing (Fig. 6f), although those human channels are clearly present at the lysosomal membrane of HEK cells (Fig. 6a-c). Does it mean that TMEM63 plays a different role in mammalian cells and that the main function of HsTMEM63 isoforms is mainly at the plasma membrane? Did you try looking at HsTMEM63 localization in Drosophila cells (S2 cells for instance) and is it also found at the lysosomal membrane? I am really not convinced by the rescue data presented in this manuscript and perhaps it should be deleted. As an alternative, I suggest to explore a similar function for TMEM63 at the lysosomes from mammalian cells. At least show a similar mechanosensitive current in isolated lysosomes of mammalian cells (for instance HEK cells) and its possible link with TMEM63 (perhaps using a combination of siRNAs).

Authors' response: Following the reviewer's suggestion, we deleted the data of the rescue experiments with *HsTMEM63s*. We also have performed lysosomal recordings to explore the functions of mammalian TMEM63. As shown in the Figures 5e and 5f of the revised manuscript, we detected similar mechanosensitive (MS) currents in the isolated lysosomes of the N2a cells, a mouse cell line. Furthermore, we found that the MS currents were largely abolished in *MsTmem63a* knock-out cells, suggesting *MsTMEM63A* mediates lysosomal mechano-sensitivity in N2a cells. In addition, overexpression of *MsTMEM63A* gave rise to a large MS currents in the lysosomal membrane. Together, these results provide direct evidence to show the conserved function of TMEM63 proteins at lysosomes.

In this revised manuscript, we have added the following new data which are relevant to the reviewer's comments: the phylogenic relationship between *MsTMEM63s* and *DmTMEM63* (Fig. 5a); the expressions of *MsTMEM63A* in N2a cells (Fig. 5b); the localization patterns of *MsTMEM63s* (Fig. 5c, d); the strategy to generate knock-out cells (Extended Data Fig. 10) and the lysosomal recording results (Fig. 5e, f). The description of the new data is in Pages 12-13, lines 259-276.

Additional points:

1) *DmTMEM63-GFP* induces mechano-sensitive cationic channels (reversing at 0 mV) recorded

in the cell-attached patch clamp configurations. This is a bit confusing for the reader as this paper concerns the role of DmTMEM63 in lysosomes. Could you please add recordings of isolated lysosomes from S2 cells (the same way as you did for the fat body cells?).

Authors' response: We appreciate the reviewer's suggestion and tried to perform lysosomal recordings of the S2 cells. Since the lysosomes in S2 cells are much smaller than those in the fat bodies, it is necessary to enlarge the lysosomes before recordings. A typical method for lysosomal enlargement is the treatment of vacuolin-1 or transfection of a constitutively-active Rab5 (Rab5^{CA}), which works well in mammalian cells³. But we found that the lysosomes in *Drosophila* S2 cells could not be enlarged using either method or even a combination of the two methods (vacuolin-1 plus Rab5^{CA}). Our observation is consistent with a previous study in which the authors stated that they could not enlarge S2 cell lysosomes suitable for patch-clamp recordings⁵. Thus, we were unable to add data of lysosomal recordings from S2 cells. We have described this technical problem in the Method section of the revised manuscript (Page 25, lines 624-625).

To avoid confusion for the readers, we also have added the following sentences in the main text "The unresponsiveness to agents for lysosomal enlargement hinders recordings of *DmTMEM63* currents from the lysosomal membrane in S2 cells. Because the fluorescent signals, although relatively weak, were also observed on the cell surface of S2 cells overexpressing *DmTMEM63*-GFP, we characterized the electrophysiological properties of plasma-membrane located *DmTMEM63* in these cells" (Page 6, lines 115-119).

2) It would be interesting to have an idea of the single channel conductance (I presume very small). Possibly a noise analysis could give you an estimate.

Authors' response: We have determined the single-channel conductance of *DmTMEM63* through noise analysis. The estimated value is very small (~1.93 pS). We have added the new data in the revised manuscript (Extended Data Fig. 3c and Page 6, lines 124).

3) Recordings of DmTMEM63 at the lysosomal membrane in fat body cells is provided and this current is suppressed in the knock out lines. What happens to the mechano-sensitive current at the plasma membrane? Do you also observe a change? (even though the plasma membrane

expression of the GFP construct seems to be very weak at the cell surface as shown in Fig. 4a and 4c., but clearly overexpression shows an obvious expression at the plasma membrane as shown in Fig. 4e and 4g).

Authors' response: In response to the reviewer's question, we have performed recordings on the plasma membranes of the fat body cells. In contrast to the mechano-sensitive (MS) currents recorded from fat-body lysosomes, we did not detect any MS currents at the plasma membranes, and there are no differences between recordings from the plasma membranes of WT and *DmTmem63*-KO cells. This result suggests that the endogenous *DmTMEM63* proteins are not localized or not functional in the plasma membrane of the fat body. We have added the new data in the revised manuscript (Extended Data Fig. 6 and Page 8, lines 167-174).

4) Can you please determine the calcium permeability of DmTMEM63? This is directly relevant to its physiological role in lysosomes.

Authors' response: In our patch-clamp recordings, we found it difficult to seal the cell membrane when the pipette solution contains high concentration of calcium ions. So we did not use electrophysiology to evaluate the calcium permeability of the channel. Instead, we performed calcium imaging. In this experiment, we directly fused GCaMP6 to the cytosolic C-terminus of *DmTMEM63*, to ensure that a putative *DmTMEM63*-permeated Ca^{2+} flux could be readily detected before diffusion and dilution in the cytosol. We treated the cells with hypo-osmotic solution which swells the cell and yields a physical stimulus involving mechanical forces⁹. We found dramatic Ca^{2+} spikes in S2 cells expressing *DmTMEM63*-fused GCaMP6, but not in cells expressing LAMP1-fused GCaMP6 (as a control), suggesting that *DmTMEM63* channel is permeable to calcium ions. Of note, we used a calcium free hypotonic solution which eliminates the extracellular Ca^{2+} influx. Thus, the Ca^{2+} signals observed in *DmTMEM63*-expressing cells are mainly derived from *DmTMEM63*-localized lysosomes.

We have added these new results in the revised manuscript. Please refer to the Extended Data Fig. 4 and Pages 6-7, lines 130-143.

5) It is mentioned in the discussion that DmTMEM63 may act as a dual osmo-mechano sensor. It

would be interesting to demonstrate experimentally that *DmTMEM63* at the lysosome membrane can also be activated by an hypo-osmotic shock.

Authors' response: As shown in the response to the point 4, we have performed additional experiments and found the *DmTMEM63* channels at lysosomes can be activated by a hypo-osmotic shock.

General comments:

A. Summary of the key results: Good

B. Originality and significance: Novel and important for the field

C. Data & methodology: validity of approach, quality of data, quality of presentation: Excellent but see the major point concerning the rescue experiments

D. Appropriate use of statistics and treatment of uncertainties: Non parametric tests should be used for small n numbers (less than 15) as you cannot be sure about the normality of data.

E. Conclusions: robustness, validity, reliability: Interesting discussion but again revise the rescue experiments

F. Suggested improvements: experiments, data for possible revision: Explore the role of mammalian TMEM33A/B/C at the lysosomal membrane of mammalian cells

G. References: appropriate credit to previous work? OK

H. Clarity and context: lucidity of abstract/summary, appropriateness of abstract, introduction and conclusions. Good but see major point

Responses to Reviewer #4 (Reviewer's comments in italics):

A. Summary of the key results

*In this manuscript Li et al., explore mechanosensing by intracellular organelles. While there has been recent excitement around the role of the nucleus as a mechanosensor for the cell, our knowledge of whether – and if so how – organelles sense and respond to force remains poorly understood. Here, they identify that the ion channel TMEM63 acts as a mechanosensor for the lysosome. Further, the authors demonstrate that the human form of TMEM63 can, at least partially, rescue function in *Tmem63* fly mutant embryos, suggesting potential conservation.*

B. Originality and significance: if not novel, please include reference

The results are important and have clear impact to cell biology. As far as I'm aware, the lysosome has not been shown to be mechanosensitive, making this an important insight.

C. Data & methodology: validity of approach, quality of data, quality of presentation

The methodology and results are generally of a high quality. I do have some suggestions for improvement below, but no major concerns.

D. Appropriate use of statistics and treatment of uncertainties

The statistical analysis is believable. Some experiments have limited replicas (e.g. Figure 5i) but these is reasonable given the long time course of this experiment.

E. Conclusions: robustness, validity, reliability

Overall, I found the results believable and well substantiated.

F. Suggested improvements: experiments, data for possible revision

I have some suggestions below.

1. Typically, modern papers have extremely dense figures with large amounts of information squeezed in. Here, Figures 1 and 2 are somewhat of an exception. The localization shown in Figure 1 has quite a lot of redundant data that can be moved to SI. I suggest merging Figures 1 and 2 into a cleaner figure that focuses on the localization of TMEM63 to lysosomes. At the moment, the additional information hinders, rather than helps, the reader in identifying the key results.

Authors' response: We thank the reviewer's constructive suggestion. In the revised manuscript, we have merged the Fig. 1 and 2 into a new Fig. 1 entitled "DmTMEM63 localizes to lysosomes". We have removed the data regarding the localization patterns of other MS channels to the Extended Data Fig. 1.

2. In general, the presentation of the microscopy imaging is poor. The contrast and colour choices (green/red) used make it hard to see specific details such as localization. Fig 4 could be improved for clarity.

Authors' response: Following the reviewer's comments, we have now adjusted the contrast and changed the color scheme from green/red to green/magenta, which makes the localization patterns more clear (please refer to the Figures 1a, 1e, 1f, 5c, 5h and Extended Data Fig. 1 of the revised manuscript). We also have improved the images for clarity in Fig. 3 of the revised manuscript.

3. In Figure 4, it is shown that the phenotypic response depends on the fed/starvation state. Yet, this is not linked back to the mechanical forces facing the cells. How are forces altered under these different conditions? For example, lysosomes are shown to change area, but is this due to a change in the mechanical environment being sensed? In brief, what role do the authors posit for mechanical forces in the observations of Figure 4?

Authors' response:

(1) How the force may be altered under fed versus starvation condition.

The mechanical environment experienced by lysosomes varies with nutrient availability. Low nutrition or starvation inhibits mTOR signaling and induces autophagy, causing lysosomes to become more active in receiving degradative cargos through fusion¹⁰ and exporting digested materials through fission¹¹ or tubulation¹². Thus, the lysosomal membrane undergoes more frequent remodeling in the starvation condition. Moreover, starvation enhances lysosomal mobility¹³, which is possibly due to the recruitment of more cytoskeletal elements that drive orientated transport of lysosomes¹⁴. Since cytoskeletal elements directly exert force on lysosomes¹⁵ and membrane remodeling involves changes in curvature and tension^{16, 17}, it is conceivable that lysosomes are subjected to more dynamic mechanical forces in the starvation conditions, and therefore, may need to be more active in the adjustment to this fast-changing mechanical environment.

(2) The link between mechanical changes and lysosomal remodeling.

With the identification of *DmTMEM63* as a lysosomal mechanosensor, it is possible that this channel mediates lysosomal Ca^{2+} efflux in response to changes in the mechanical state of the lysosomal membrane. Our calcium imaging experiment included in the revised manuscript (Extended Data Fig. 4) supports the notion that *DmTMEM63* activation results in lysosomal Ca^{2+}

efflux. Lysosomal Ca^{2+} release in turn regulates lysosomal fusion, fission and mobility which are essential for morphological remodeling and normal function of lysosomes. Thus, we hypothesize that the lysosome remodels its morphology in accordance with the mechanical forces and this mechanical-to-morphological coupling requires *Dm*TMEM63. A disruption of the coupling results in an abnormal lysosomal morphology and positioning (Fig. 3); starvation boosts the role of mechanical forces in shaping lysosomal morphology, thus causing a more severe phenotypes than those under fed condition (Fig. 3).

In the Discussion section of the revised manuscript, we have added sentences to describe the mechanical force experienced by the lysosome and its relation with the cellular phenotypes observed. Please refer to Page 14, lines 299-306.

4. From a conceptual view (and on a similar theme to the above point), it is not clear that the phenotypes observed in Figure 5 are due to a loss of mechanosensitivity. TMEM63 may be acting through multiple pathways. Can larvae be raised under different environment stress conditions and their adult viability/climbing be assessed? One may imagine that the Tmem63 mutants would have even lower climbing index if derived from larvae that grew in a more stressed environment (as compared to changes in wt).

Authors' response: We have performed new experiments following the reviewer's suggestion. We raised the larvae in starvation condition and found that, compared to WT, the *Tmem63* mutant flies showed reduced viability. The mutant flies also exhibited climbing defects at day 4 of adulthood, a stage in which mutant flies normally performed well under fed conditions. These results suggest that the *Tmem63* mutant flies are vulnerable to starvation and the nutritional stress accelerates the manifestation of the behavioral defects. We have added the new data in the revised manuscript (Extended Data Fig. 9 and Page 10-11, lines 228-232).

5. The authors should provide at least a hypothesis for why HsTMEM63A rescues (partially) whereas B and C variants do not. Is there something about the specific conserved domains that gives insight into the action of TMEM63 in Drosophila?

Authors' response: In general, TMEM63 proteins all consist of 11 transmembrane domains (from TM0 to TM10) with quite similar structures⁸, suggesting a common ion-conducting mechanism for the channel itself. In order to search for possible variations among TMEM63 homologues, we considered the non-transmembrane domains which are important in modulating channel activity through interaction with TM domains or other cellular factors¹⁸. According to the topology⁸, the TMEM63 proteins have two major cytosolic domains, one is a linker between TM2 and TM3 (termed as the intracellular linker, IL) and the other is a C-terminal tail (termed as C-terminal domain, CTD). We found through alignment that the length of the IL domain is comparable among three members of TMEM63 protein, while the CTD is variable among different TMEM63s. As shown in the Figure below, the CTD is much shorter in *Dm*TMEM63 than that in the mammalian TMEM63B or TMEM63C, and the length of the CTD in TMEM63A is in between. Given that *Dm*TMEM63 has full rescue effect, *Hs*TMEM63B/C has no effect and *Hs*TMEM63A has partial effect, the length of the CTD seems to be correlated with the actions of these proteins in *Drosophila*. Since the C-terminus of TMEM63 resides in the cytosol and may be a modulation site through interaction with other cellular factors as well as TM domains, we hypothesize that the long C-terminal tails in *Hs*TMEM63B/C proteins may hinder the potential regulatory effect which are essential for actions of TMEM63 in *Drosophila*.

Protein	IL length	CTD length	Rescue effect
Dm TMEM63	161	72	Full rescue
Hs TMEM63A	184	86	Partial rescue
Ms TMEM63A	183	85	ND
Hs TMEM63B	183	99	No rescue
Ms TMEM63B	182	99	ND
Hs TMEM63C	181	97	No rescue
Ms TMEM63C	181	96	ND

Table legend: The length of the IL and CTD domains of TMEM63 homologs, and their rescue effect on the behavior of the *Tem63* mutant fly. The length is counted as the number of the residues. ND, not determined.

In this revision, we have removed the data regarding the ectopic rescue experiment with *HsTMEM63s* (as suggested by Reviewer #3). Therefore, we just raised this hypothesis in the response letter but did not add it in the revised manuscript.

6. In the Discussion, the authors highlight that Ca dynamics should be recorded in conjunction with lysosome membrane remodelling. Though a full characterisation is not required, it does seem a clear omission (particularly given the prevalence of good Ca reporters) that such imaging is not performed here. It would clearly help link the purported changes in local mechanics (as the membrane changes) with action of TMEM63.

Authors' response: Following the reviewer's suggestion, we have performed an additional experiment to examine the correlation between mechanical changes and the actions of TMEM63. We transfected S2 cells with a tandem fluorescence-tagged *DmTMEM63* (*DmTMEM63*-mCherry-GCaMP6f), which enables us to simultaneously visualize lysosomal morphology (mCherry signal) and monitor the *DmTMEM63*-permeated Ca^{2+} flux (GCaMP6 signal). We found the lysosomal Ca^{2+} signals are highly dynamic in the native cellular environment. We then segmented lysosomal membranes, calculated the membrane curvature of each segment (as a representation of the local mechanics) and quantified its Ca^{2+} intensity (as a representation of the *DmTMEM63* activity). We found a positive correlation between the Ca^{2+} signal intensity and the membrane curvature, suggesting the action of *DmTMEM63* channel is in accordance with the morphological remodeling of lysosomes. We have added the new data in the revised manuscript (Extended Data Fig. 7a-c and Pages 8-9, lines 177-187).

We also have noticed that, although the correlation is positive and significant, it is not in a high value (the averaged correlation coefficient is 0.16 ± 0.03). This is possibly due to the spatial limitation of the imaging system in which the local curvature of a lysosome can not be determined precisely. In the Discussion section of the revised manuscript, we pointed out this issue and proposed that a live-imaging system with super-resolution microscopy may be warranted in future studies to assess the relation between TMEM63 action and lysosomal dynamics. Please refer to Page 14, lines 308-313.

G. Clarity and context: lucidity of abstract/summary, appropriateness of abstract, introduction and conclusions

The paper is well written and the results are clear.

References

1. Rusten, T.E. *et al.* Programmed Autophagy in the Drosophila Fat Body Is Induced by Ecdysone through Regulation of the PI3K Pathway. *Developmental Cell* **7**, 179-192 (2004).
2. Scott, R.C., Schuldiner, O. & Neufeld, T.P. Role and Regulation of Starvation-Induced Autophagy in the Drosophila Fat Body. *Developmental Cell* **7**, 167-178 (2004).
3. Chen, C.-C. *et al.* Patch-clamp technique to characterize ion channels in enlarged individual endolysosomes. *Nature Protocols* **12**, 1639-1658 (2017).
4. Braulke, T. & Bonifacino, J.S. Sorting of lysosomal proteins. *Biochimica et Biophysica Acta (BBA) - Molecular Cell Research* **1793**, 605-614 (2009).
5. Wie, J. *et al.* A growth-factor-activated lysosomal K⁺ channel regulates Parkinson's pathology. *Nature* **591**, 431-437 (2021).
6. del Marmol, J.I., Touhara, K.K., Croft, G. & MacKinnon, R. Piezo1 forms a slowly-inactivating mechanosensory channel in mouse embryonic stem cells. *eLife* **7**, e33149 (2018).
7. Shi, J. *et al.* Sphingomyelinase Disables Inactivation in Endogenous PIEZO1 Channels. *Cell Reports* **33** (2020).
8. Zheng, W. *et al.* TMEM63 proteins function as monomeric high-threshold mechanosensitive ion channels. *Neuron* (2023).
9. Roffay, C. *et al.* Passive coupling of membrane tension and cell volume during active response of cells to osmosis. *Proceedings of the National Academy of Sciences* **118**, e2103228118 (2021).
10. Luzio, J.P., Pryor, P.R. & Bright, N.A. Lysosomes: fusion and function. *Nature Reviews Molecular Cell Biology* **8**, 622-632 (2007).
11. Saffi, G.T. & Botelho, R.J. Lysosome Fission: Planning for an Exit. *Trends in Cell Biology* **29**, 635-646 (2019).
12. Du, W. *et al.* Kinesin 1 Drives Autolysosome Tubulation. *Developmental Cell* **37**, 326-336 (2016).
13. Korolchuk, V.I. *et al.* Lysosomal positioning coordinates cellular nutrient responses. *Nature Cell Biology* **13**, 453-460 (2011).
14. Kast, D.J. & Dominguez, R. The Cytoskeleton-Autophagy Connection. *Current Biology* **27**, R318-R326 (2017).
15. Cason, S.E. & Holzbaur, E.L.F. Selective motor activation in organelle transport along axons. *Nature Reviews Molecular Cell Biology* **23**, 699-714 (2022).
16. Rangamani, P., Mandadap, Kranthi K. & Oster, G. Protein-Induced Membrane Curvature Alters Local Membrane Tension. *Biophysical Journal* **107**, 751-762 (2014).
17. Lipowsky, R. Spontaneous tubulation of membranes and vesicles reveals membrane tension generated by spontaneous curvature. *Faraday Discussions* **161**, 305-331 (2013).
18. Barros, F., Domínguez, P. & de la Peña, P. Cytoplasmic domains and voltage-dependent potassium channel gating. *Frontiers in pharmacology* **3**, 49 (2012).

Decision Letter, first revision:

Our ref: NCB-A51032A

30th November 2023

Dear Dr. Jan,

Thank you very much for submitting your revised manuscript, "The Drosophila TMEM63 channel is a lysosomal mechanosensor" (NCB-A51032A). It has been evaluated again by our original Reviewers #1-2-4, and Rev#1 helped us assess your responses to Reviewer #3 as well. Their comments are below. The reviewers find the conclusions stronger after revision, and I am pleased to say that in light of their comments, we shall be happy, in principle, to publish your manuscript in Nature Cell Biology pending minor revisions to comply with our editorial and formatting guidelines.

Please note that the current version of your manuscript is in a PDF format. Could you please email us a copy of the file in an editable format (Microsoft Word or LaTeX) as we can not proceed with PDFs at this stage? Thank you in advance for your attention to this point.

With the Word file in-hand, we will be performing detailed checks on your paper and will send you a checklist detailing our editorial and formatting requirements in about 1-2 weeks. Please do not upload the final materials and make any revisions until you receive this additional information from us.

Thank you again for your interest in Nature Cell Biology. Please do not hesitate to contact me if you have any questions.

Sincerely,

Melina

Melina Casadio, PhD
Senior Editor, Nature Cell Biology
ORCID ID: <https://orcid.org/0000-0003-2389-2243>

Reviewer #1 (Remarks to the Author):

The reviewers have satisfactorily addressed all my previous comments. The major comment I had was related to the function of human TMEM63s. The author now nicely show that at least one of them also forms a mechano-sensitive channel in lysosomes.

The paper reveals a novel mechano-sensing ion channel on an intracellular organelle. It also reveals some of the channel's important physiological function.

Reviewer #2 (Remarks to the Author):

The authors have satisfactorily addressed all my comments. The rebuttal letter was also beautifully written, as was the manuscript. This is a major breakthrough in the lysosomal physiology field. Congratulations to the authors!

Reviewer #4 (Remarks to the Author):

The authors have done an excellent job of dealing with my concerns. The conceptual novelty is better supported and the image presentation much improved. Reading through the reports, it looks like they have addressed all the major concerns.

Author Rebuttal, first revision:

Responses to Reviewer #1 (Reviewer's comments in italics):

Remarks to the Author:

The reviewers have satisfactorily addressed all my previous comments. The major comment I had was related to the function of human TMEM63s. The author now nicely show that at least one of them also forms a mechano-sensitive channel in lysosomes.

The paper reveals a novel mechano-sensing ion channel on an intracellular organelle. It also reveals some of the channel's important physiological function.

Authors' response: Thanks very much for the reviewer's positive comments. We appreciate the reviewer's insightful suggestions which have helped us to strengthen the manuscript.

Responses to Reviewer #2 (Reviewer's comments in italics):

Remarks to the Author:

The authors have satisfactorily addressed all my comments. The rebuttal letter was also beautifully written, as was the manuscript. This is a major breakthrough in the lysosomal physiology field. Congratulations to the authors!

Authors' response: We thank the reviewer for the very helpful suggestions which have guided us to perform additional experiments and analysis to improve the manuscript.

Responses to Reviewer #4 (Reviewer's comments in italics):

Remarks to the Author:

The authors have done an excellent job of dealing with my concerns. The conceptual novelty is better supported and the image presentation much improved. Reading through the reports, it looks like they have addressed all the major concerns.

Authors' response: We sincerely appreciate the reviewer's supportive comments. Thanks very much for the reviewer's valuable suggestions.

Decision Letter, second revision:

Our ref: NCB-A51032A

8th December 2023

Dear Dr. Jan,

Thank you for your patience as we've prepared the guidelines for final submission of your Nature Cell Biology manuscript, "The Drosophila TMEM63 channel is a lysosomal mechanosensor" (NCB-A51032A). Please carefully follow the step-by-step instructions provided in the attached file, and add a response in each row of the table to indicate the changes that you have made. Please also check and comment on any additional marked-up edits we have proposed within the text. Ensuring that each point is addressed will help to ensure that your revised manuscript can be swiftly handed over to our production team.

If you have not done so already, please alert us to any related manuscripts from your group that are under consideration or in press at other journals, or are being written up for submission to other

journals (see: <https://www.nature.com/nature-research/editorial-policies/plagiarism#policy-on-duplicate-publication> for details).

In recognition of the time and expertise our reviewers provide to Nature Cell Biology's editorial process, we would like to formally acknowledge their contribution to the external peer review of your manuscript entitled "The *Drosophila* TMEM63 channel is a lysosomal mechanosensor". For those reviewers who give their assent, we will be publishing their names alongside the published article.

Nature Cell Biology offers a Transparent Peer Review option for new original research manuscripts submitted after December 1st, 2019. As part of this initiative, we encourage our authors to support increased transparency into the peer review process by agreeing to have the reviewer comments, author rebuttal letters, and editorial decision letters published as a Supplementary item. When you submit your final files please clearly state in your cover letter whether or not you would like to participate in this initiative. Please note that failure to state your preference will result in delays in accepting your manuscript for publication.

Cover suggestions

COVER ARTWORK: We welcome submissions of artwork for consideration for our cover. For more information, please see our guide for cover artwork.

Nature Cell Biology has now transitioned to a unified Rights Collection system which will allow our Author Services team to quickly and easily collect the rights and permissions required to publish your work. Approximately 10 days after your paper is formally accepted, you will receive an email in providing you with a link to complete the grant of rights. If your paper is eligible for Open Access, our Author Services team will also be in touch regarding any additional information that may be required to arrange payment for your article.

Please note that *Nature Cell Biology* is a Transformative Journal (TJ). Authors may publish their research with us through the traditional subscription access route or make their paper immediately open access through payment of an article-processing charge (APC). Authors will not be required to make a final decision about access to their article until it has been accepted. Find out more about Transformative Journals

For information regarding our different publishing models please see our Transformative Journals page. If you have any questions about costs, Open Access requirements, or our legal forms,

please contact ASJournals@springernature.com.

Please use the following link for uploading these materials:
[Redacted]

Best regards,

Kendra Donahue
Staff
Nature Cell Biology

On behalf of

Melina Casadio, PhD
Senior Editor, Nature Cell Biology
ORCID ID: <https://orcid.org/0000-0003-2389-2243>

Reviewer #1:

Remarks to the Author:

The reviewers have satisfactorily addressed all my previous comments. The major comment I had was related to the function of human TMEM63s. The author now nicely show that at least one of them also forms a mechano-sensitive channel in lysosomes.

The paper reveals a novel mechano-sensing ion channel on an intracellular organelle. It also reveals some of the channel's important physiological function.

Reviewer #2:

Remarks to the Author:

The authors have satisfactorily addressed all my comments. The rebuttal letter was also beautifully written, as was the manuscript. This is a major breakthrough in the lysosomal physiology field. Congratulations to the authors!

Reviewer #4:

Remarks to the Author:

The authors have done an excellent job of dealing with my concerns. The conceptual novelty is better supported and the image presentation much improved. Reading through the reports, it looks like they have addressed all the major concerns.

Final Decision Letter:

Dear Dr Jan,

I am pleased to inform you that your manuscript, "Drosophila TMEM63 and mouse TMEM63A are lysosomal mechanosensory ion channels", has now been accepted for publication in Nature Cell Biology.

Please note that *Nature Cell Biology* is a Transformative Journal (TJ). Authors may publish their research with us through the traditional subscription access route or make their paper immediately open access through payment of an article-processing charge (APC). Authors will not be required to make a final decision about access to their article until it has been accepted. Find out more about Transformative Journals

If you have not already done so, we strongly recommend that you upload the step-by-step protocols used in this manuscript to the Protocol Exchange (www.nature.com/protocolexchange), an open online resource established by Nature Protocols that allows researchers to share their detailed experimental know-how. All uploaded protocols are made freely available, assigned DOIs for ease of citation and are fully searchable through nature.com. Protocols and Nature Portfolio journal papers in which they are used can be linked to one another, and this link is clearly and prominently visible in the online versions of both papers. Authors who performed the specific experiments can act as primary authors for the Protocol as they will be best placed to share the methodology details, but the Corresponding Author of the present research paper should be included as one of the authors. By uploading your Protocols to Protocol Exchange, you are enabling researchers to more readily reproduce or adapt the methodology you use, as well as increasing the visibility of your protocols and papers. You can also establish a dedicated page to collect your lab Protocols. Further information can be found at www.nature.com/protocolexchange/about

With kind regards,

Melina Casadio, PhD
Senior Editor, Nature Cell Biology
ORCID ID: <https://orcid.org/0000-0003-2389-2243>
